# Behavioral control by depolarized and hyperpolarized states of an integrating neuron

Aylesse Sordillo[1], Cornelia I Bargmann[1,2]*

[1]Lulu and Anthony Wang Laboratory of Neural Circuits and Behavior, The Rockefeller University, New York, United States; [2]Chan Zuckerberg Initiative, Redwood City, United States

**Abstract** Coordinated transitions between mutually exclusive motor states are central to behavioral decisions. During locomotion, the nematode *Caenorhabditis elegans* spontaneously cycles between forward runs, reversals, and turns with complex but predictable dynamics. Here, we provide insight into these dynamics by demonstrating how RIM interneurons, which are active during reversals, act in two modes to stabilize both forward runs and reversals. By systematically quantifying the roles of RIM outputs during spontaneous behavior, we show that RIM lengthens reversals when depolarized through glutamate and tyramine neurotransmitters and lengthens forward runs when hyperpolarized through its gap junctions. RIM is not merely silent upon hyperpolarization: RIM gap junctions actively reinforce a hyperpolarized state of the reversal circuit. Additionally, the combined outputs of chemical synapses and gap junctions from RIM regulate forward-to-reversal transitions. Our results indicate that multiple classes of RIM synapses create behavioral inertia during spontaneous locomotion.

**\*For correspondence:**
cori@rockefeller.edu

**Competing interest:** The authors declare that no competing interests exist.

## Introduction

Neurons coordinate their activity across networks using a variety of signals: fast chemical transmitters, biogenic amines, neuropeptides, and electrical coupling via gap junctions (*Tritsch and Sabatini, 2012*; *Zell et al., 2020*; *Taylor et al., 2019*; *Liu et al., 2017*; *Nagy et al., 2019*). Signals from many neurons coalesce to generate large-scale brain activity patterns that are correlated with movement, while reflecting the animal's memory, internal state, and sensory experience (*Kato et al., 2015*; *Musall et al., 2019*). The mechanisms for generating stable, mutually exclusive activity and behavioral states across networks, while allowing behavioral flexibility, are incompletely understood.

The relationships between neurons, synapses, circuits, and behavior can be addressed precisely in the compact nervous system of *Caenorhabditis elegans*. Like many animals, *C. elegans* has locomotion-coupled, global brain activity states (*Kato et al., 2015*; *Musall et al., 2019*; *Nguyen et al., 2016*; *Venkatachalam et al., 2016*). Many of its integrating interneurons and motor neurons are active during one or more of three basic motor behaviors – forward runs, reversals, and turns (*Figure 1A*). A set of interneurons including AIB, AVA, and RIM are active when animals reverse (*Gordus et al., 2015*; *Kato et al., 2015*; *Nguyen et al., 2016*; *Venkatachalam et al., 2016*); a different set, AIY, RIB, and AVB, are active during forward runs (*Kaplan et al., 2020*; *Kato et al., 2015*; *Li et al., 2014*; *Nguyen et al., 2016*); and a set including AIB, RIB, and RIV are active during sharp omega turns, which typically follow a reversal (*Kato et al., 2015*; *Nguyen et al., 2016*; *Venkatachalam et al., 2016*; *Wang et al., 2020*). The functional role of each integrating neuron can be evaluated by considering the neuron's regulation of specific locomotor features, like reversal speed or turn angle, and its influence on locomotor transitions. The AVA neurons, for example, are backward command neurons

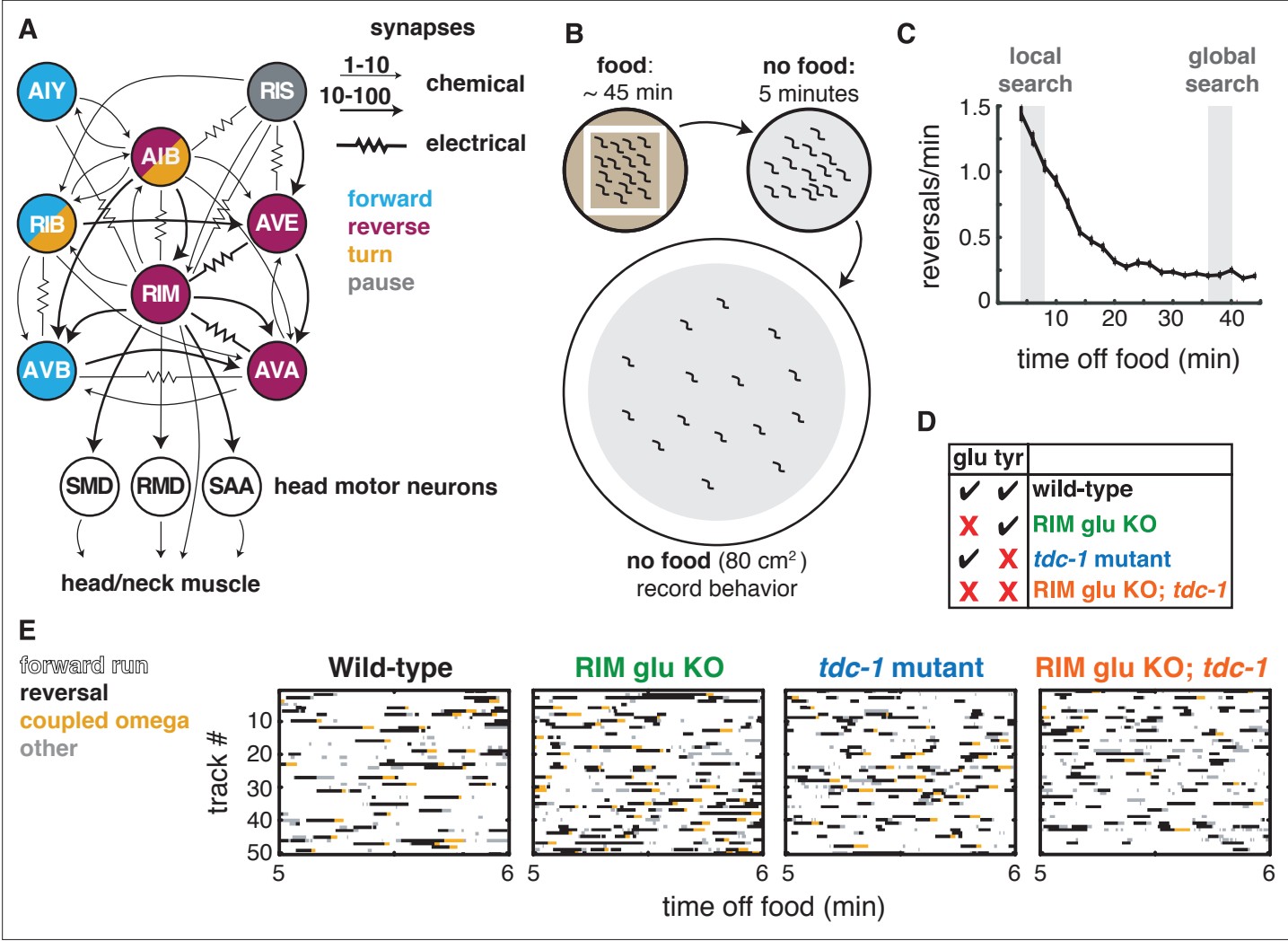

**Figure 1.** Two RIM neurotransmitters affect spontaneous locomotion. (**A**) RIM synapses with interneurons, motor neurons, and muscle implicated in spontaneous foraging behavior (*Cook et al., 2019*; *White et al., 1986*). Colors of neurons indicate associated locomotor states based on neural manipulations and functional calcium imaging (*Alkema et al., 2005*; *Gray et al., 2005*; *Kato et al., 2015*; *Li et al., 2014*; *Pokala et al., 2014*; *Steuer Costa et al., 2019*; *Wang et al., 2020*; *Zheng et al., 1999*). (**B**) Off-food foraging assay. (**C**) Mean reversals per minute of wild-type animals in foraging assays. Vertical lines indicate standard error of the mean. Gray shaded boxes indicate local search (4–8 min off food) and global search (36–40 min off food) intervals analyzed in subsequent figures. n = 324. (**D**) RIM neurotransmitter mutants. RIM glu KO: RIM-specific knockout of the vesicular glutamate transporter EAT-4 (*Figure 1—figure supplement 1*). *tdc-1*, tyrosine decarboxylase mutant, which lacks tyramine in RIM and octopamine in RIC. (**E**) Ethograms of 50 randomly chosen tracks per genotype during minute 5–6 of local search. Color code: white, forward runs; black, reversals; yellow, omega turns coupled to a reversal; gray, pauses, shallow turns, and omega turns that were not preceded by a reversal.

The online version of this article includes the following figure supplement(s) for figure 1:

**Figure supplement 1.** CRISPR-Cas9-generated alleles enable RIM-specific glutamate transporter knockout.

that drive reversals; when AVA neurons are optogenetically depolarized, animals reverse, and when AVA neurons are ablated or acutely silenced reversals are short and infrequent (*Chalfie et al., 1985*; *Gordus et al., 2015*; *Liu et al., 2017*; *Pokala et al., 2014*; *Roberts et al., 2016*). Acute silencing of AVA often causes aberrant pauses – thwarted reversals – followed by a turn, indicating that AVA neurons are required for the execution of a reversal, but not for the global dynamics of the forward-reversal-turn sequence (*Kato et al., 2015*; *Pokala et al., 2014*). Other neurons in the locomotor circuit are implicated in transition dynamics. For example, altering AIB and RIB activity can change the probability and timing of the reversal-to-turn transition without generating abnormal pause states (*Pokala et al., 2014*; *Wang et al., 2020*).

Among the interneurons in the locomotor circuit, RIM, a pair of motor/interneurons, has both straightforward and apparently paradoxical functions (*Figure 1A*). RIM is active during both spontaneous and stimulus-evoked reversals, and its activity correlates with reversal speed (*Gordus et al., 2015*; *Kagawa-Nagamura et al., 2018*; *Kato et al., 2015*). RIM releases the neurotransmitter tyramine, which extends reversals by inhibiting the AVB forward-active neurons and suppresses head oscillations by inhibiting the head muscles, in both cases through the tyramine-gated chloride channel LGC-55 (*Alkema et al., 2005*; *Pirri et al., 2009*). RIM tyramine also sharpens reversal-coupled omega turns by activating SER-2, a G protein-coupled receptor on motor neurons (*Donnelly et al., 2013*). In addition to these effects on locomotion parameters, RIM has puzzling effects on behavioral transitions. Optogenetic depolarization of RIM drives reversals, but ablation of RIM paradoxically increases spontaneous reversals, indicating that RIM can either induce or suppress reversals (*Gordus et al., 2015*; *Gray et al., 2005*; *Guo et al., 2009*; *López-Cruz et al., 2019*; *Zheng et al., 1999*). RIM also mediates competition between sensory inputs and motor circuits, generating variability in behavioral responses to external stimuli (*Gordus et al., 2015*; *Ji et al., 2019*), and biases choices between attractive and aversive stimuli (*Ghosh et al., 2016*; *Hapiak et al., 2013*; *Li et al., 2012*; *Wragg et al., 2007*). On longer timescales, RIM modulates learning as well as physiological responses to temperature or unfolded protein stress (*De Rosa et al., 2019*; *Fu et al., 2018*; *Ha et al., 2010*; *Jin et al., 2016*; *Özbey et al., 2020*).

Here, we develop an integrated view of RIM's role in locomotor features, motor transitions, and behavioral dynamics through cell-specific manipulation of its synapses. In addition to tyramine, RIM expresses the vesicular glutamate transporter EAT-4, identifying it as one of the 38 classes of glutamatergic neurons in *C. elegans* (*Lee et al., 1999*; *Serrano-Saiz et al., 2013*). RIM also forms gap junctions with multiple neurons whose activity is associated with reversals (AIB, AVA, AVE), as well as neurons active during pauses (RIS) and forward runs (AIY) (*Cook et al., 2019*; *White et al., 1986*; *Figure 1A*). *eat-4* and gap junction subunits are broadly expressed throughout the foraging circuit, precluding a simple interpretation of null mutants in these genes (*Bhattacharya et al., 2019*; *Serrano-Saiz et al., 2013*). Therefore, we used a cell-specific knockout of *eat-4* and a cell-specific gap junction knockdown to isolate the synaptic functions of RIM. By examining behavioral effects of multiple transmitters and gap junctions, we reveal distinct functions of RIM during reversals, when its activity is high, and during forward locomotion, when its activity is low. Notably, our results indicate that while RIM depolarization extends reversals, the propagation of hyperpolarization through RIM gap junctions extends the opposing forward motor state. This work indicates that a single interneuron class employs different classes of synapses to shape mutually exclusive behaviors.

## Results

### RIM glutamate and tyramine suppress spontaneous reversals and increase reversal length

The goal of this work was to understand how RIM influences spontaneous behavioral dynamics, including individual features of locomotion and transitions between motor states. We used an off-food foraging assay in which forward, reversal, and turn behaviors emerge from predictable internal states (*Calhoun et al., 2014*; *Gray et al., 2005*; *Hills et al., 2004*; *López-Cruz et al., 2019*; *Wakabayashi et al., 2004*; *Figure 1B*). When removed from food and placed on a featureless agar surface, *C. elegans* engages in local search, in which a high frequency of spontaneous reversals limits dispersal from the recently encountered food source. Over about 15 min, animals spontaneously transition into global search, a state with infrequent reversals and long forward runs that promotes dispersal (*Figure 1C*). We recorded animals throughout this assay, and identified and quantified reversals, turns, forward runs, and pauses from behavioral sequences (example tracks in *Figure 1E*). The full dataset is available for further analysis (*Source data 1*, Dryad, GitHub, see Materials and methods), and a summary of results is included in Figure 8.

We began by examining the effects of RIM glutamate on local search (*Figure 1D and E*). *C. elegans* mutants lacking the vesicular glutamate transporter *eat-4* or various glutamate receptors have abnormal local search behaviors (*Baidya et al., 2014*; *Chalasani et al., 2007*; *Choi et al., 2015*; *Hills et al., 2004*; *López-Cruz et al., 2019*). To selectively inactivate glutamatergic transmission from RIM, we used an FRT-flanked endogenous *eat-4* locus and expressed FLP recombinase under a *tdc-1*

promoter, which intersects with *eat-4* only in RIM (*López-Cruz et al., 2019*; *Figure 1—figure supplement 1*). The resulting animals lacking RIM glutamate had an increased frequency of reversals during local search, but not global search (*Figures 1E and 2A and B*). To ensure that these effects were caused by the desired mutation, we examined controls with the modified *eat-4* locus alone or the FLP recombinase alone, accompanied by the same fluorescent marker (*Figure 2—figure supplement 1*). For all experiments performed here, the 'wild-type' (WT) controls include appropriate genetic controls and transgenic marker controls; for full genotypes, see *Supplementary file 1*, Table 1: Strain details.

RIM is the primary neuronal source of tyramine, whose synthesis requires the tyrosine decarboxylase encoded by *tdc-1* (*Alkema et al., 2005*). As previously reported, *tdc-1* mutants had an increased reversal frequency during local search (*Figures 1E and 2A and B*; *Alkema et al., 2005*). *tdc-1* is also expressed in RIC neurons, where it is used, together with *tbh-1*, in the biosynthesis of the neurotransmitter octopamine (*Alkema et al., 2005*). *tbh-1* did not affect reversal frequency during local search, identifying tyramine as the relevant transmitter for reversals (*Figure 2—figure supplement 2*). The RIM glu KO; *tdc-1* double mutant was similar to each single mutant (*Figure 2A and B*). Thus, both of RIM's neurotransmitters, glutamate and tyramine, suppress spontaneous reversals.

Reversals during local search segregate into short reversals of less than half a body length, and long reversals that average >1 body length (*Gray et al., 2005*; *Figure 2C and D*). Using these criteria, both short and long reversals increased in frequency in RIM glu KO animals during local search, but only short reversals increased in frequency in *tdc-1* mutants or the RIM glu KO; *tdc-1* double mutant (*Figure 2D*). To better understand this distinction, we conducted an analysis of the full reversal length distribution (*Figure 2E*). In fact, both RIM glu KO animals and *tdc-1* mutants had decreased reversal lengths compared to WT, with a stronger effect in *tdc-1* mutants, indicating that RIM glutamate and tyramine both extend reversal length.

Long reversals are more likely to be followed by an omega turn than short reversals (*Chalasani et al., 2007*; *Croll, 2009*; *Gray et al., 2005*; *Huang et al., 2006*; *Wang et al., 2020*; *Zhao et al., 2003*; *Figure 2F*). The fraction of reversal-omega maneuvers was reduced in *tdc-1* mutants (*Figure 2G*, left), while pure reversals that terminate in an immediate forward run increased (*Figure 2G*, right); as previously reported, omega turn angles were also shallower in *tdc-1* mutants (*Figure 2H*). RIM glu KO animals had normal reversal-omega frequencies and turn angles after reversals, despite a decrease in reversal length (*Figure 2G and H*).

## RIM neurotransmitters distinguish reversal and reversal-omega behaviors

Analysis of the frequency distributions of all reversal lengths, speeds, and durations uncovered additional distinctions between the functions of RIM glutamate and tyramine (*Figure 3A–I*). First, while reversal lengths were decreased in a graded fashion by RIM glu KO or *tdc-1* (*Figures 2E and 3A*), reversal speeds were substantially reduced only in *tdc-1* mutants (*Figure 3B*). *tdc-1* and RIM glu KO had similar decreases in reversal durations (*Figure 3C*). We found that genetic markers and background controls could affect these parameters by up to 12%; with that in mind, we discuss only effect sizes of ≥0.15 in these and other quantitative experiments (see Materials and methods and *Figure 2—figure supplement 1*).

Separating different classes of reversals (*Figure 2F and G*) revealed that the RIM glu KO decreased reversal-omega duration but did not affect pure reversal duration (*Figure 3D–I*). *tdc-1* mutants decreased the duration of reversal-omegas, increased the duration of pure reversals, and decreased the speed of all reversals (*Figure 3D–I*). RIM glu KO; *tdc-1* double mutant animals resembled *tdc-1* mutants. These results are in agreement with previous work showing that pure reversals and reversal-omega maneuvers have distinct kinetics and circuit requirements (*Wang et al., 2020*).

Forward runs are heterogeneous compared to reversals, with an exponential distribution of durations (*Figure 2—figure supplement 2*; *Wakabayashi et al., 2004*). Neither RIM glu KO animals nor *tdc-1* mutants had strong effects on forward run durations compared to controls (*Figure 2—figure supplement 2*). Both *tdc-1* and *tbh-1* mutants had substantially diminished forward speeds, suggesting a role of octopamine in forward locomotion (*Figure 2—figure supplement 2*). Because the octopaminergic RIC neurons were not the focus of this work, forward speed was not examined further.

In summary, tyramine affects both the speed and the duration of all classes of reversals, whereas RIM glutamate only increases the duration of reversals that are coupled to omega turns. RIM

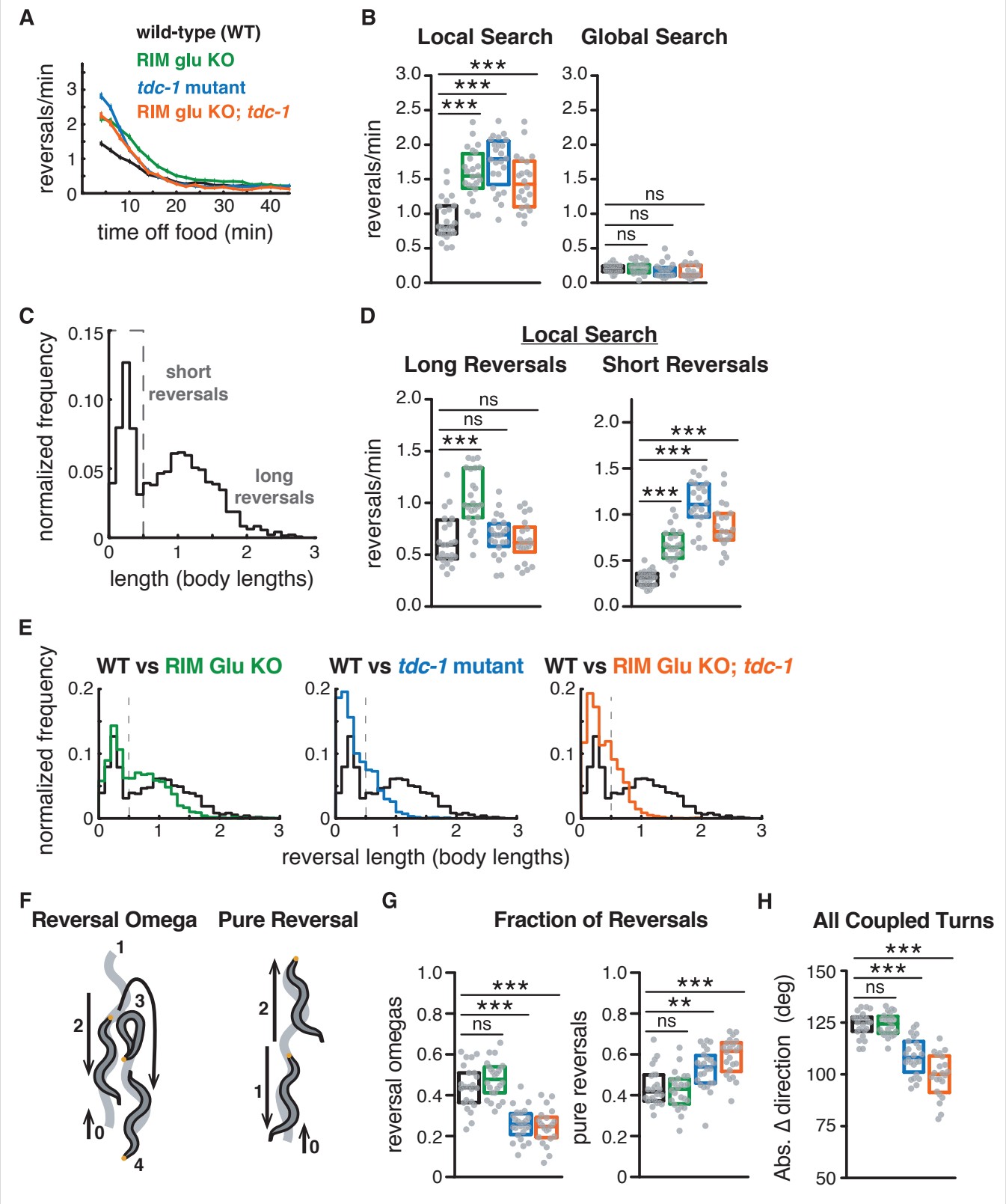

**Figure 2.** RIM glutamate and tyramine suppress spontaneous reversals and increase reversal length. (**A**) Mean reversals per minute in foraging assays for genotypes analyzed in *Figures 2–3*. Vertical dashes indicate standard error of the mean. n = 296–332. All strains bear *tdc-1p*::nFLP and the *elt-2p*::nGFP marker (*Figure 2—figure supplement 1*, *Supplementary file 1*, Table 3: Reversals and forward runs, n values). (**B**) Mean frequency of all reversals during local search (4–8 min off food, left) and global search (36–40 min off food, right). (**C**) Normalized probability distribution of wild-type reversal

*Figure 2 continued on next page*

*Figure 2 continued*

lengths during local search. Short reversals cover less than 0.5 body lengths. (**D**) Mean frequency of long reversals (>0.5 body lengths, left) and short reversals (<0.5 body lengths, right) during local search. (**E**) Normalized probability distribution of mutant reversal lengths during local search, plotted with WT distributions. (**F**) A forward-moving animal (0) initiates a reversal (1–2) that is coupled to an omega turn (3) and terminates in forward movement (4) (reversal-omega, left). A forward-moving animal (0) initiates a reversal (1) that terminates in forward movement (2) (pure reversal, right). Yellow dot indicates nose. (**G**) Fraction of all reversals during local search that terminate in an omega turn (left) or forward movement (right) for each genotype. (**H**) Absolute change in direction after a reversal-turn maneuver (including omega and shallower turns) for each genotype. (**B, D, G, H**) Each gray dot is the mean for 12–15 animals on a single assay plate (**Source data 1**), with 22–24 plates per genotype. Boxes indicate median and interquartile range for all assays. Asterisks indicate statistical significance compared to WT using a Kruskal–Wallis test with Dunn's multiple comparisons test (**p-value<0.01, ***p-value<0.001, ns = p-value≥0.05). (**C, E**) n = 1443–2760 events per genotype. The reversal defects in (RIM) tyramine- and (RIC) octopamine-deficient *tdc-1* mutants are not shared by octopamine-deficient *tbh-1* mutants (**Figure 2—figure supplement 2**).

The online version of this article includes the following figure supplement(s) for figure 2:

**Figure supplement 1.** RIM::nFLP and edited *eat-4* do not account for RIM glu KO phenotype.

**Figure supplement 2.** Octopamine affects forward and reversal speed, but not reversal length or frequency.

neurotransmitters do not substantially affect forward run durations, consistent with low RIM activity during forward runs. RIC octopamine increases forward and reversal speed.

## Additional RIM transmitters contribute to global search dynamics

In addition to glutamate and tyramine, RIM expresses multiple neuropeptides (*flp-18, pdf-2*, and others) (*Bhardwaj et al., 2018*; *Ghosh et al., 2016*; *Taylor et al., 2019*). Release of both classical transmitters and neuropeptides is inhibited by the tetanus toxin light chain, which cleaves the synaptic vesicle fusion protein synaptobrevin (*Schiavo et al., 1992*). Expression of tetanus toxin in RIM and RIC under the *tdc-1* promoter resulted in defects resembling those of *tdc-1* mutants (*Figure 4A–C*): reversal frequency increased, while reversal length, speed, and durations decreased, during local search behavior (*Figure 4D–G*). Efficient RIM-only promoters are not available, but expression of tetanus toxin in RIC alone caused only minor defects in reversal frequency and speed, implicating RIM as a major regulator of reversal parameters (*Figure 4—figure supplement 1*, *Figure 4—figure supplement 2*). RIC tetanus toxin expression reduced forward locomotion speed to a similar extent as *tbh-1* mutants (*Figure 4—figure supplement 1*).

The expression of tetanus toxin in RIM and RIC also increased reversals during the global search period, an effect that was not observed in RIM glutamate KO or tyramine-deficient mutants (*Figure 4A and C*). Tetanus toxin expression in RIC alone did not affect reversal frequency during global search (*Figure 4—figure supplement 1*). These results suggest that another transmitter from RIM, perhaps a neuropeptide, suppresses reversals during global search.

## Artificial hyperpolarization of RIM reveals unexpected functions in forward runs

To relate RIM functions to its membrane potential, we hyperpolarized RIM by expressing the *Drosophila* histamine-gated chloride channel (HisCl) under the *tdc-1* promoter and exposing the animals to histamine while off food (*Pokala et al., 2014*; *Figure 5A and B*). Unexpectedly, silencing RIM with HisCl led to a substantial decrease in spontaneous reversal frequency, which was most evident during local search (*Figure 5A–C*). The effects on reversal frequency were opposite to those of RIM ablation, RIM neurotransmitter mutants, or RIM::tetanus toxin expression, all of which increased spontaneous reversal frequency (*Alkema et al., 2005*; *Gray et al., 2005*; *Figures 2–4*). The decrease in reversals was accompanied by an increase in forward run durations (*Figure 5D*). The opposite effects of RIM ablation and acute silencing suggest that RIM has active functions when hyperpolarized that stabilize and extend forward runs.

Reversal length, speed, and duration were greatly reduced by hyperpolarization of RIM, effects that were similar to but stronger than the effect of *tdc-1* or tetanus toxin (*Figure 5E*, *Figure 5—figure supplement 1*). These results suggest that RIM glutamate and tyramine are released when RIM is depolarized, as expected, to extend reversals and increase reversal speed. However, the stronger effects of RIM::HisCl indicate that hyperpolarization affects other targets as well.

A possible explanation for the distinct effects of RIM::HisCl silencing with histamine, versus RIM inactivation with mutations or ablation, is that acute and chronic neuronal silencing have different

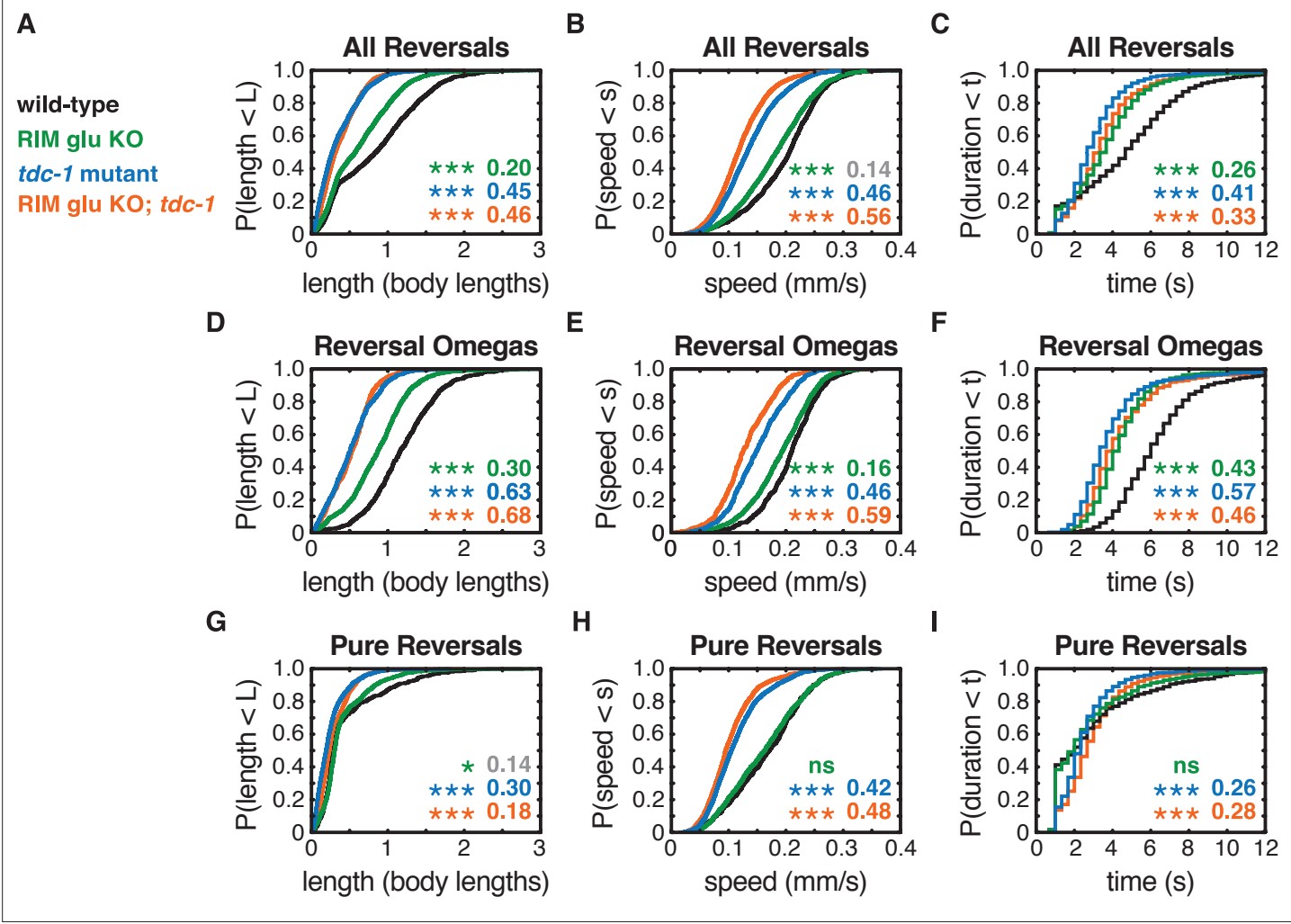

**Figure 3.** RIM neurotransmitters differently affect pure reversal and reversal-omega behaviors. (A–C) For all reversals during local search, empirical cumulative distributions of reversal length (A), reversal speed (B), and reversal duration (C). (D–F) For reversal-omega maneuvers during local search, empirical cumulative distributions of reversal length (D), reversal speed (E), and reversal duration (F). (G–I) For pure reversals during local search, empirical cumulative distributions of reversal length (G), reversal speed (H), and reversal duration (I). Asterisks indicate statistical significance compared to WT using a two-sample Kolmogorov–Smirnov test, with a Bonferroni correction (*p-value<0.05, ***p-value<0.0001, ns = p-value≥0.05). Numbers in figures indicate effect size (see Materials and methods). Although statistically significant, the smaller effect sizes indicated in gray are similar to values from control strains (e.g., *Figure 2—figure supplement 1*) and fall under the 0.15 cutoff for discussion established from those controls. n = 500–3132 events from 22 to 24 assays, 12–15 animals per assay (*Supplementary file 1*, Table 3: Reversals and forward runs, n values). The strong reversal defects in (RIM) tyramine- and (RIC) octopamine-deficient *tdc-1* mutants are not shared by octopamine-deficient *tbh-1* mutants, which do affect forward locomotion (*Figure 2—figure supplement 2*).

effects (*Yeon et al., 2021*). To examine this possibility, we incubated RIM::HisCl animals on histamine for 48 hr, beginning in the L2 larval stage, and tested their behavior as adults. Chronic histamine treatment caused decreases in reversal frequency, reversal length, and reversal speed that were similar to those in acutely treated animals (*Figure 5F–H*, *Figure 5—figure supplement 2*).

We hypothesized that RIM hyperpolarization might suppress reversals by decreasing the activity of the AVA backward command neurons. In order to test this possibility, we examined spontaneous calcium dynamics in AVA after hyperpolarizing RIM::HisCl with histamine. Immobilized animals were exposed to *Escherichia coli*-conditioned media, then switched to buffer to induce a local search-like state characterized by sustained epochs of high AVA activity (Figure 5—figure supplement 3). Acute RIM silencing decreased the fraction of time that AVA activity was high and reduced spontaneous AVA transitions from the low- to the high-activity state (*Figure 5—figure supplement 3*). AVA activity fell after 30 min in buffer, consistent with a change to a global search-like state; at the same time,

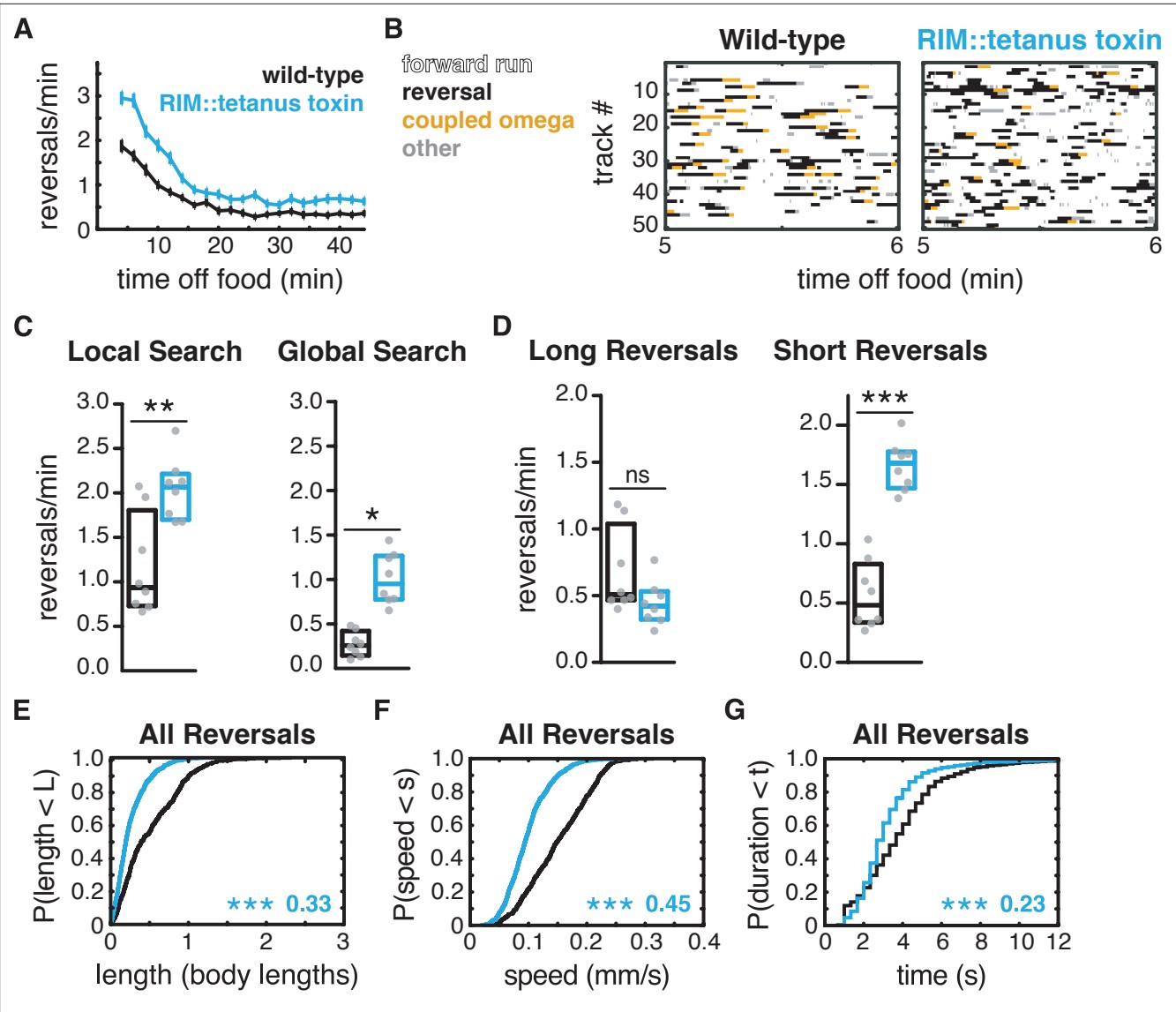

**Figure 4.** Additional RIM transmitters contribute to global search dynamics. (**A**) Mean reversals per minute in animals expressing tetanus toxin light chain under the RIM- and RIC-specific *tdc-1* promoter. Reversal defects are milder or absent when tetanus toxin is expressed under the RIC-specific *tbh-1* promoter (**Figure 4—figure supplement 1**). Vertical dashes indicate standard error of the mean. n = 103–111. (**B**) Ethograms of 50 randomly chosen tracks per genotype during minute 5–6 of local search. Color code: white, forward runs; black, reversals; yellow, omega turns coupled to a reversal; gray, pauses, shallow turns, and omega turns that were not preceded by a reversal. (**C**) Mean frequency of all reversals during local search (4–8 min off food, left) and global search (36–40 min off food, right). (**D**) Mean frequency of long reversals (>0.5 body lengths, left) and short reversals (<0.5 body lengths, right) during local search. (**E–G**) For all reversals during local search, empirical cumulative distributions of reversal length (**E**), reversal speed (**F**), and reversal duration (**G**). (**C, D**) Each gray dot is the mean for 12–15 animals on a single assay plate (**Source data 1**), with eight plates per genotype. Boxes indicate median and interquartile range for all assays. Asterisks indicate statistical significance compared to WT using a Mann–Whitney test (*p-value<0.05, **p-value<0.01, ***p-value<0.001, ns = p-value≥0.05). (**E–G**) Asterisks indicate statistical significance compared to WT using a two-sample Kolmogorov–Smirnov test (***p-value<0.0001). Numbers indicate effect size. n = 595–1066 events from eight assays, 12–15 animals per assay (**Supplementary file 1**, Table 3: Reversals and forward runs, n values).

The online version of this article includes the following figure supplement(s) for figure 4:

**Figure supplement 1.** Expression of tetanus toxin in RIC affects forward and reversal speed, with small effects on reversal frequency.

**Figure supplement 2.** Expression of tetanus toxin in RIM decreases reversal-omega coupling and alters reversal-omega and pure reversal duration.

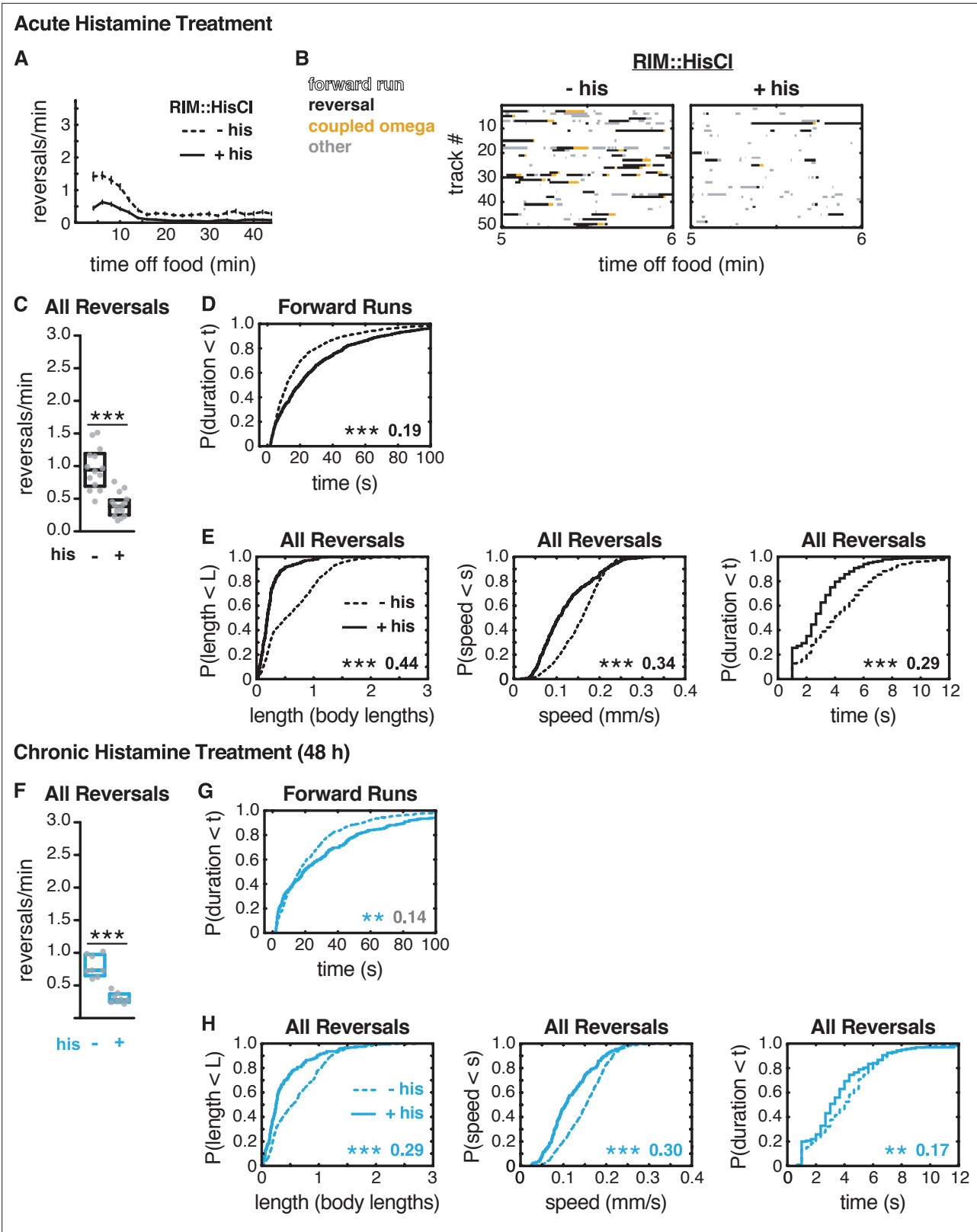

**Figure 5.** Artificial hyperpolarization of RIM extends forward runs and suppresses reversals. (**A–E**) Acute hyperpolarization of RIM::HisCl with histamine. (**F–H**) Chronic hyperpolarization of RIM::HisCl with histamine (48 hr). (**A**) Mean reversals per minute in animals expressing HisCl in RIM, with (+his) or without (–his) acute histamine treatment. Vertical dashes indicate standard error of the mean. n = 93–109. (**B**) Ethograms of 50 randomly chosen tracks per genotype during minute 5–6 of local search. Color code: white, forward runs; black, reversals; yellow, omega turns coupled to a reversal;

*Figure 5 continued on next page*

*Figure 5 continued*

gray, pauses, shallow turns, and omega turns that were not preceded by a reversal. (**C, F**) Mean frequency of all reversals during local search, with or without histamine, in RIM::HisCl animals. Each gray dot is the mean for 12–15 animals on a single assay plate (*Source data 1*), with 14–16 plates per genotype. Boxes indicate median and interquartile range for all assays. Asterisks indicate statistical significance compared to untreated controls using a Mann–Whitney test (***p-value<0.001). (**D, G**) Durations of forward runs during local search with and without histamine, in RIM::HisCl animals; empirical cumulative distributions include all runs ≥ 2 s. (**E, H**) For all reversals during local search, empirical cumulative distributions of reversal length, reversal speed, and reversal duration, with (solid lines) and without (dashed lines) histamine, in RIM::HIsCl animals. (**D, E, G, H**) Asterisks indicate statistical significance compared to untreated controls using a two-sample Kolmogorov–Smirnov test (**p-value<0.01, ***p-value<0.0001). Numbers indicate effect size. Although statistically significant, the smaller effect size indicated in gray is similar to values from control strains (e.g., *Figure 2—figure supplement 1*). n = 394–1071 events per genotype from 14 to 17 assays (**D, E**), n = 156–487 events per genotype from eight assays (**G, H**), 12–15 animals per assay (*Supplementary file 1*, Table 3: Reversals and forward runs, n values).

The online version of this article includes the following figure supplement(s) for figure 5:

**Figure supplement 1.** Artificial hyperpolarization of RIM in *tdc-1* mutants.

**Figure supplement 2.** Behavioral recovery in RIM::HisCl animals after histamine removal.

**Figure supplement 3.** RIM hyperpolarization decreases AVA activity in paralyzed animals during a local search-like state.

the effects of RIM silencing were diminished. These results are consistent with a model in which RIM hyperpolarization acutely suppresses AVA activity. However, as neuronal dynamics in immobilized and freely moving animals are substantially different, they may not fully reflect the effects of RIM hyperpolarization on AVA during off-food foraging (*Hallinen et al., 2021*).

## RIM gap junctions stabilize forward runs

To explain the effect of hyperpolarized RIM neurons, we considered the gap junctions that RIM forms with a variety of other neurons in the local search circuit (*Figure 1A*). RIM shares the most gap junctions with AVA and AVE that, like RIM, have high activity during reversals and low activity during forward runs. We hypothesized that RIM gap junctions stabilize the forward motor state by propagating hyperpolarizing currents between RIM and AVA (and possibly other) neurons, thereby preventing their depolarization.

Invertebrate gap junctions are made up of innexin subunits, which assemble as homo- or heteromers of eight subunits on each of the two connected cells (*Burendei et al., 2020*; *Oshima et al., 2016*). Most *C. elegans* neurons express multiple innexin genes; RIM neurons express 11 innexin genes, including *unc-9* (*Bhattacharya et al., 2019*). *unc-9* is expressed in many classes of neurons, and mutants have strong defects in forward and backward locomotion (*Brenner, 1974*; *Kawano et al., 2011*; *Liu et al., 2017*; *Liu et al., 2006*; *Park and Horvitz, 1986*; *Shui et al., 2020*; *Starich et al., 2009*). To bypass its broad effects, neuronal *unc-9* function can be reduced in a cell-specific fashion by expressing *unc-1(n494),* a dominant negative allele of a stomatin-like protein that is an essential component of neuronal UNC-9 gap junctions (*Chen et al., 2007*; *Jang et al., 2017*). We knocked down UNC-9 gap junctions in RIM by driving *unc-1(n494)* cDNA under the *tdc-1* promoter. While unlikely to inactivate all RIM innexins and gap junctions, this manipulation should alter *unc-9* gap junction signaling in a RIM-selective manner.

RIM gap junction knockdown animals had superficially coordinated locomotion and exhibited the characteristic shift from local to global search over time (*Figure 6A*). However, these gap junction knockdown animals had a greatly increased frequency of reversals compared to WT (*Figure 6A–C*). Both short and long reversals were increased in frequency during both local search and global search, while reversal length, speed, and duration were only slightly reduced (*Figure 6C–G*, *Figure 6—figure supplement 1*). The RIM gap junction knockdown also resulted in a substantial decrease in forward run duration (*Figure 6H*, *Figure 6—figure supplement 1*).

Combining the gap junction knockdown with a *tdc-1* mutation yielded additive effects, with both forward and reversal parameters altered (*Figure 6—figure supplement 2*). Knocking down UNC-9 gap junctions in RIC alone did not affect reversal frequency, but decreased both forward speed and reversal speed, as well as forward run duration (*Figure 6—figure supplement 3*). These results support the hypothesis that *unc-9*-containing gap junctions in RIM promote forward locomotion.

To ask whether the *unc-9*-containing gap junctions propagate the effects of RIM hyperpolarization, we crossed the RIM gap junction knockdown into the RIM::HisCl strain. Combining the RIM gap junction knockdown with RIM hyperpolarization resulted in mutual suppression of their effect on reversal

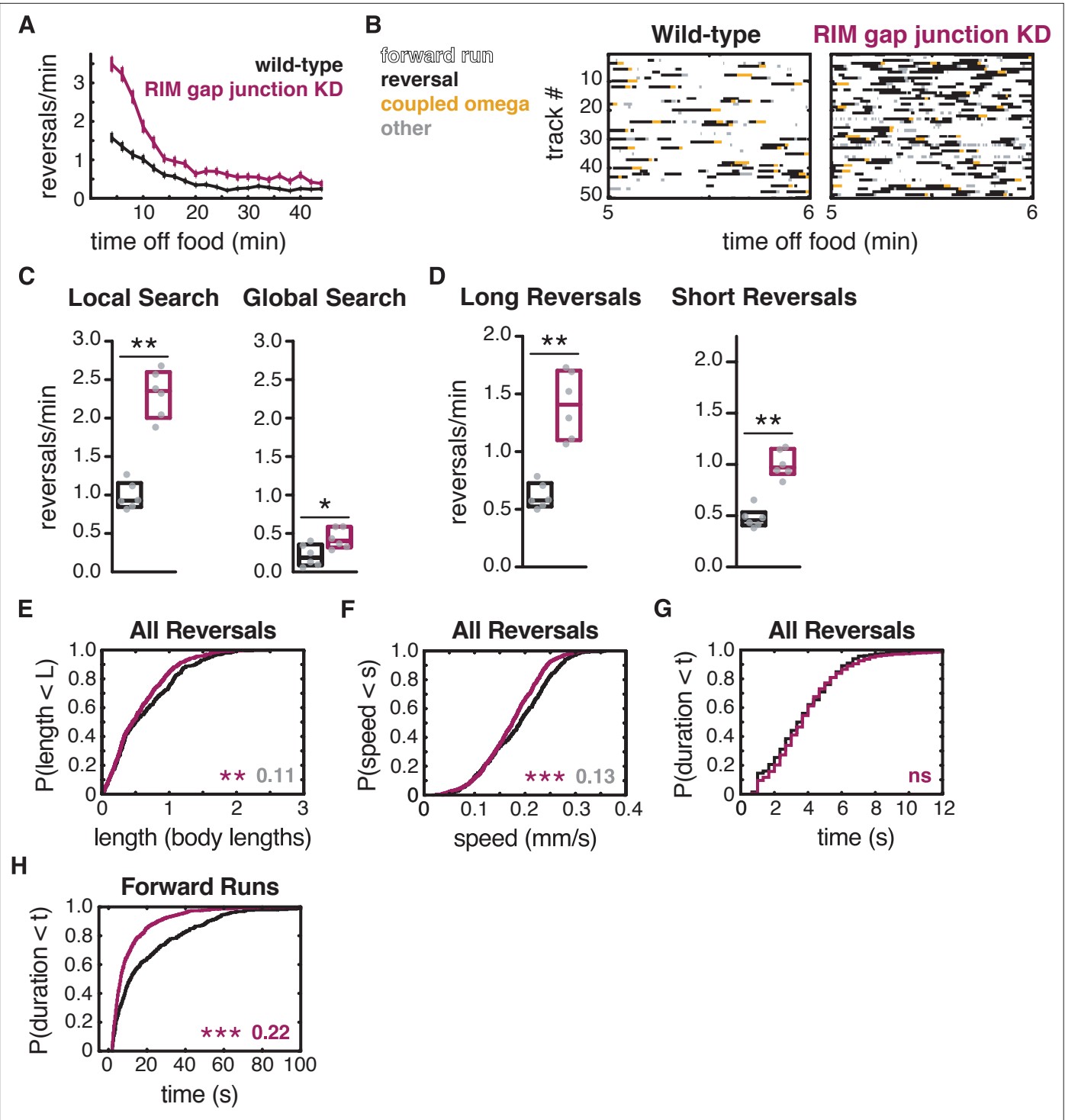

**Figure 6.** RIM gap junctions suppress spontaneous reversals. (**A**) Mean reversals per minute in animals bearing an *unc-1(n494)* dominant negative transgene to knock down *unc-9*-containing gap junctions (RIM gap junction KD). Vertical dashes indicate standard error of the mean. n = 77–85. (**B**) Ethograms of 50 randomly chosen tracks per genotype during minute 5–6 of local search. Color code: white, forward runs; black, reversals; yellow, omega turns coupled to a reversal; gray, pauses, shallow turns, and omega turns that were not preceded by a reversal. (**C**) Mean frequency of all reversals during local search (4–8 min off food, left) and global search (36–40 min off food, right). (**D**) Mean frequency of long reversals (>0.5 body lengths, left) and short reversals (<0.5 body lengths, right) during local search. (**E–G**) For all reversals during local search, empirical cumulative distributions of reversal length (**E**), reversal speed (**F**), and reversal duration (**G**). (**H**) Durations of forward runs during local search; empirical cumulative distributions include all runs ≥ 2 s. (**C, D**) Each gray dot is the mean for 12–15 animals on a single assay plate (*Source data 1*), with six assays per genotype. Boxes indicate median and interquartile range for all assays. Asterisks indicate statistical significance compared to WT using a Mann–Whitney

*Figure 6 continued on next page*

*Figure 6 continued*

test (*p-value<0.05, **p-value<0.01). (**E–H**) Asterisks indicate statistical significance compared to WT using a two-sample Kolmogorov–Smirnov test (**p-value<0.01, ***p-value<0.001, ns = p-value≥0.05). Numbers indicate effect size. Although statistically significant, the smaller effect sizes indicated in gray fall below the 0.15 cutoff for discussion established by observing control strains (e.g., *Figure 2—figure supplement 1*). n = 330–933 events per genotype from six assays, 12–15 animals per assay (*Supplementary file 1*, Table 3: Reversals and forward runs, n values). Note that the *tdc-1* promoter also expresses *unc-1(n494)* in RIC. Reversal frequencies are not altered in an RIC-selective *unc-1(n494)* strain, but forward and reversal speed and forward run duration are decreased (*Figure 6—figure supplement 3*).

The online version of this article includes the following figure supplement(s) for figure 6:

**Figure supplement 1.** RIM gap junctions affect reversal-omega frequency.

**Figure supplement 2.** RIM gap junctions and RIM tyramine act additively in spontaneous local search behavior.

**Figure supplement 3.** RIC gap junctions affect forward and reversal speed, but not reversal frequency.

frequency, so that double mutants had a similar reversal frequency to wild-type animals (*Figure 7A and B*). The shortened forward run durations observed in RIM gap junction knockdown animals were also suppressed when RIM was hyperpolarized (*Figure 7C*). These results suggest that forward states are stabilized in part, but not entirely, through *unc-9*-containing gap junctions.

## Strong depolarization of RIM engages neurotransmitter-independent functions

Optogenetic depolarization of RIM rapidly increases reversal frequency (*Gordus et al., 2015*; *Guo et al., 2009*; *López-Cruz et al., 2019*; *Figure 7D*). The frequency of optogenetically induced reversals was unaffected by *tdc-1*, RIM glu KO, or the double mutant, whether examined during local search or during global search (*Figure 7D*, *Figure 7—figure supplement 1*). This result suggests that RIM does not require tyramine or glutamate neurotransmitters to trigger optogenetically induced reversals.

We considered whether RIM gap junctions might propagate optogenetic depolarization to AVA command neurons. The RIM gap junction knockdown did not affect optogenetically induced reversal frequencies during local search, but it did decrease optogenetically induced reversals during global search (*Figure 7F*, *Figure 7—figure supplement 1*). These results suggest a minor role for *unc-9* gap junctions in the initiation of optogenetically induced reversals.

Optogenetically induced reversals were shorter in RIM glu KO, tyramine-deficient, and RIM gap junction knockdown animals than in WT (*Figure 7E and G*). Thus, optogenetically induced reversals are extended by all RIM synaptic outputs.

## Discussion

A cycle of forward runs interrupted by reversals and turns dominates the spontaneous locomotion of *C. elegans* during local search. We show here that RIM generates behavioral inertia to inform the dynamics of these locomotor states (*Figure 8*). RIM stabilizes reversals through its chemical synapses while depolarized and stabilizes forward runs through its gap junctions while hyperpolarized. Together with other results (*Kawano et al., 2011*), our results suggest that hyperpolarization through gap junctions is a recurrent circuit motif in *C. elegans* locomotion.

## RIM neurotransmitters cooperate to stabilize reversals

RIM controls specific locomotor features: it increases spontaneous reversal speed and duration (*Gray et al., 2005*, this work), suppresses head oscillations during reversals, and sharpens the omega turns coupled to reversals (*Alkema et al., 2005*; *Donnelly et al., 2013*; *Pirri et al., 2009*). These functions all rely on the RIM transmitter tyramine, which also increases the length of reversals evoked by aversive sensory stimuli (*Alkema et al., 2005*; *Pirri et al., 2009*). We found that RIM glutamate increases spontaneous reversal length and duration, but only during the coupled reversal-omega maneuver, and does not increase reversal speed. Both RIM glutamate and tyramine also extend reversals evoked by acute depolarization.

Neurons that release both classical transmitters, like glutamate, and biogenic amines, like tyramine, can employ them additively, cooperatively, or distinctly. In mice, dopaminergic neurons that project from the ventral tegmental area to the nucleus accumbens release both dopamine and glutamate, and

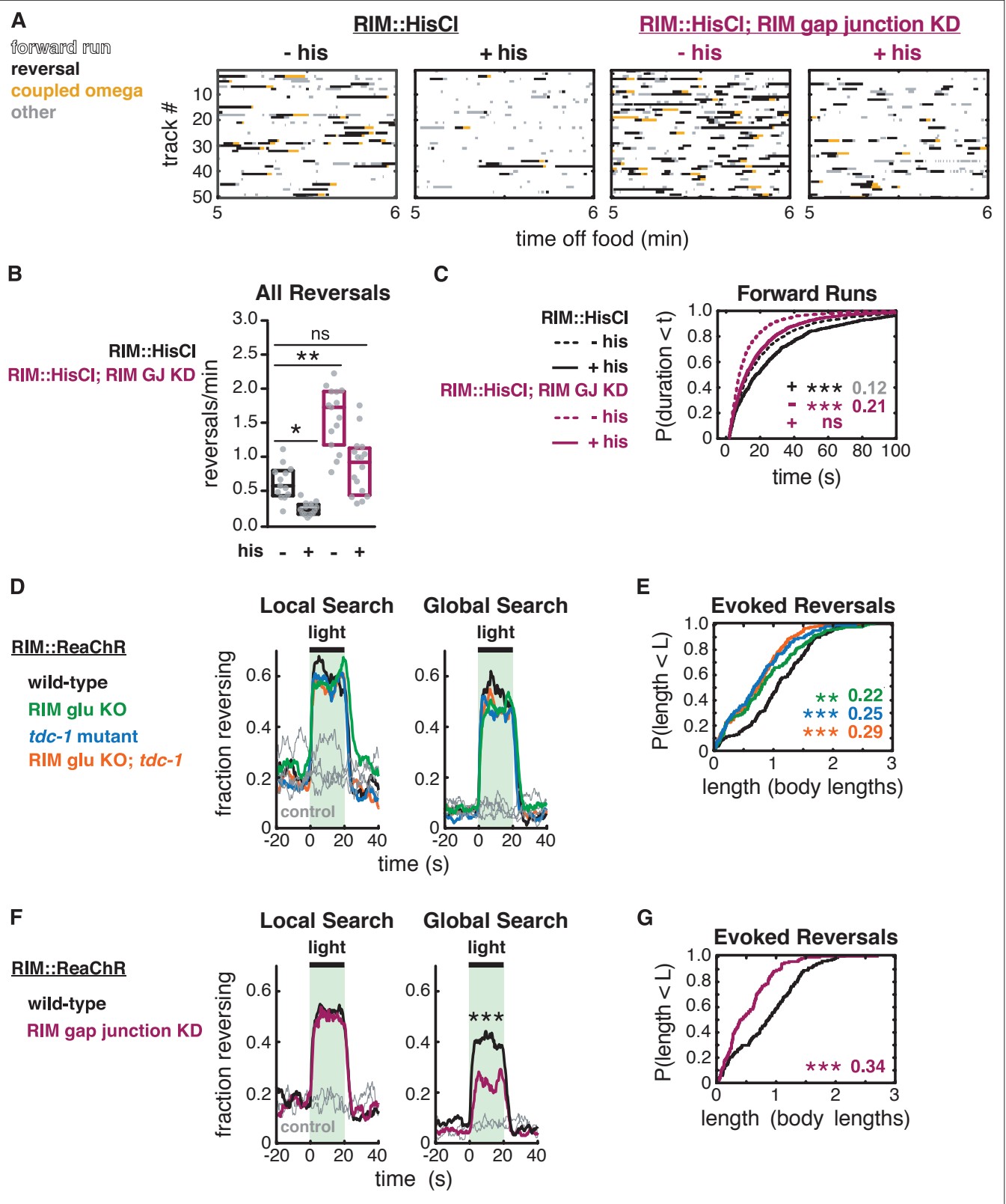

**Figure 7.** RIM gap junctions mediate effects of hyperpolarization and depolarization. (**A–C**) Behavior of RIM::HisCl and RIM::HisCl; RIM gap junction knockdown animals. (**A**) Ethograms of 50 randomly chosen tracks per genotype during minute 5–6 of local search. Color code: white, forward runs; black, reversals; yellow, omega turns coupled to a reversal; gray, pauses, shallow turns, and omega turns that were not preceded by a reversal. (**B**) Mean frequency of all reversals during local search (4–8 min off food), with or without histamine. Each gray dot is the mean for 12–15 animals on a single

*Figure 7 continued on next page*

*Figure 7 continued*

assay plate (**Source data 1**), with 13–16 plates per genotype. Boxes indicate median and interquartile range for all assays. Asterisks indicate statistical significance compared to untreated WT controls using a Kruskal–Wallis test with Dunn's multiple comparisons test (*p-value<0.05, **p-value<0.01, ns = p-value≥0.05). (**C**) Durations of forward runs during local search with (solid lines) and without (dashed lines) histamine; empirical cumulative distributions include all runs ≥2 s. n = 768–1994 events from 13 to 16 assays, 12–15 animals per assay (**Supplementary file 1**, Table 3: Reversals and forward runs, n values). (**D–G**) Effects of RIM::ReaChR activation in wild-type, RIM glu KO, *tdc-1* mutants, RIM glu KO; *tdc-1* double mutants, and RIM gap junction knockdown animals. (**D, F**) Animals were exposed to light for 20 s (green shading), with or without all-trans retinal pretreatment, during local search (8–14 min off food, left) or global search (38–44 min off food, right). Neurotransmitter mutants do not suppress optogenetically evoked reversals (**D**). RIM gap junction knockdown suppresses optogenetically evoked reversals during global search (**F**) (***p<0.001, **Figure 7—figure supplement 1**). Similar results were obtained at lower and higher light levels. (**E, G**) For all reversals induced during the light pulse during local search (8–14 min off food), empirical cumulative distributions of reversal length. All animals were pretreated with all-trans retinal. n = 119–193 reversals from 12 to 15 assays, 12–15 animals per assay, 2 (**E**) or 3 (**G**) light pulses per assay conducted 8–14 min after removal from food (**Supplementary file 1**, Table 3: Reversals and forward runs, n values). (**C, E, G**) Asterisks indicate statistical significance compared to controls of the same genotype using a two-sample Kolmogorov–Smirnov test with a Bonferroni correction (**p-value<0.01, ***p-value<0.001, ns = p-value≥0.05). Numbers indicate effect size.

The online version of this article includes the following figure supplement(s) for figure 7:

**Figure supplement 1.** RIM gap junctions support optogenetically evoked reversals.

either transmitter can support positive reinforcement (**Zell et al., 2020**). In both *Drosophila* and mice, the glutamate transporter enhances dopamine loading into synaptic vesicles for a cooperative effect

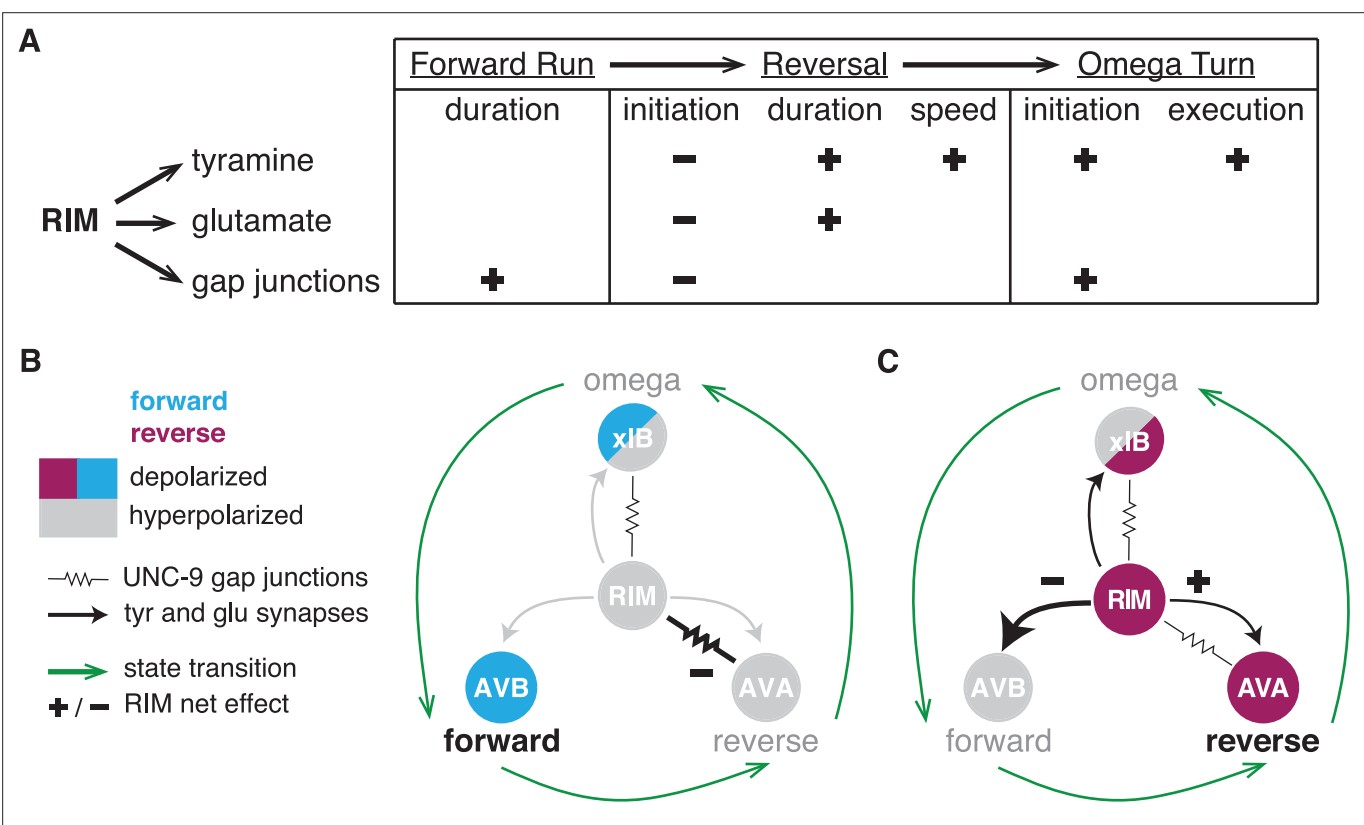

**Figure 8.** RIM synapses generate behavioral inertia and jointly regulate forward-to-reversal transitions. (**A**) Summary of synaptic regulation of spontaneous behaviors (**Figures 2–7**). (+) indicates that the normal function of the synapse increases the behavioral parameter (e.g., RIM tyramine increases reversal speed because *tdc-1* mutant reversals are slower than wild-type). (–) indicates that the synapse decreases the parameter (e.g., RIM tyramine, glutamate, and *unc-1/unc-9* gap junctions all inhibit reversal initiation because the mutants have more spontaneous reversals than wild-type). Additional RIM transmitters inhibit reversals during global search (**Figure 4**), and RIC octopamine and gap junctions increase forward and reversal speed, but not reversal initiation (**Figure 2—figure supplement 2**, **Figure 4—figure supplement 1**, **Figure 6—figure supplement 3**). (**B, C**) AVB, AVA, and xIB (AIB + RIB) are representative of the neurons that promote forward runs, reversals, and omega turns, respectively. AVB and RIB are depolarized during forward runs; RIM, AVA, and AIB are depolarized during reversals; AIB and RIB are depolarized during turns. (**B**) RIM *unc-1/unc-9* gap junctions stabilize forward runs by propagating a hyperpolarizing signal to reversal-promoting neurons. (**C**) RIM tyramine and glutamate stabilize reversals by inhibiting forward-promoting neurons and may also activate reversal-promoting neurons.

(*Aguilar et al., 2017*; *Münster-Wandowski et al., 2016*). At a more subtle level, GABA and dopamine co-released from terminals in the mammalian striatum affect target neurons differently – GABA rapidly inhibits action potentials, while dopamine modulates activity through slower GPCR pathways (*Tritsch and Sabatini, 2012*). The nonadditive effects of RIM glutamate and tyramine on spontaneous and optogenetically evoked behavioral dynamics in *C. elegans* suggest that they act as co-transmitters to cooperatively stabilize reversal states when RIM is depolarized.

*C. elegans* glutamate receptors and tyramine receptors are broadly expressed in the locomotor circuit. Among RIM's synaptic targets, the AIB and AVA reversal-promoting neurons express excitatory AMPA-type glutamate receptors, as does RIM itself (*Brockie et al., 2001*; *Hart et al., 1995*; *Taylor et al., 2019*), and glutamate is released onto AVA during reversals (*Marvin et al., 2013*). RIM glutamate might reinforce the reversal state by depolarizing AVA, while RIM tyramine inhibits the competing forward state via the tyramine-gated chloride channel LGC-55 on AVB (*Pirri et al., 2009*). AVA also expresses a glutamate-gated chloride channel, *avr-14,* that inhibits spontaneous reversals; a genetic interaction between *avr-14* and an RIM-specific knockdown of *eat-4* suggests that this receptor could be a target of RIM glutamate (*Li et al., 2020*). Since all of these receptors coexist in a circuit rich in positive and negative feedback (*Roberts et al., 2016*), cell-specific knockouts of receptors as well as neurotransmitters may be needed to define their functions precisely.

The promoter used to alter RIM activity and signaling, *tdc-1,* is also expressed in RIC neurons. RIC-specific manipulation did not appreciably affect reversal frequency or duration, although it might contribute alongside RIM. RIC neurotransmitters and gap junctions did affect both forward and reversal speed.

## RIM gap junctions extend forward runs

For both chemogenetic hyperpolarization and optogenetic depolarization of RIM, the effects on reversal frequency were opposite to those predicted from RIM ablation. Hyperpolarization led to an unanticipated increase in forward run durations, pointing to an active function for RIM when silenced. RIM and its gap junction partners AVA, AVE, and AIB have low activity during forward locomotion; our results suggest that in the hyperpolarized forward state RIM gap junctions inhibit the AVA backward command neurons and possibly others as well (*Gordus et al., 2015*; *Kagawa-Nagamura et al., 2018*; *Kato et al., 2015*).

Our conclusion that RIM gap junctions stabilize a hyperpolarized state resonates with previous studies in a different part of the reversal circuit. In addition to their gap junctions with RIM, the AVA neurons form gap junctions with *unc-9*-expressing VA and DA motor neurons that drive backward locomotion. Genetic inactivation of those gap junctions results in defects in forward locomotion and increases calcium levels in AVA (*Kawano et al., 2011*). From this result, the UNC-9-UNC-7 gap junctions were inferred to decrease the activity of AVA based on hyperpolarizing current flow from VA and DA motor neurons. This role is similar to the role we propose for gap junctions between AVA and RIM. In fact, it could be molecularly similar: the *unc-9* innexin expressed in VA/DA neurons and RIM can form heterotypic gap junctions with the *unc-7* innexin expressed in AVA (*Kawano et al., 2011*). However, unlike the RIM gap junction knockdown, which acts primarily to affect the duration of coordinated forward runs, the AVA-motor neuron knockdown results in highly uncoordinated movement.

The experiments here, and in *Kawano et al., 2011*, are limited by the fact that behavior and calcium imaging do not directly measure gap junction conductances. Moreover, direct measurements of gap junctions between AVA and VA5 motor neurons indicate that UNC-9-UNC-7 gap junctions transmit depolarizing current from VA5 to AVA (*Liu et al., 2017*; *Shui et al., 2020*). That said, reconstitution in *Xenopus* oocytes revealed a startling array of rectifying and membrane potential-dependent properties of UNC-9-UNC-7 gap junctions, depending on which of the seven UNC-7 splice forms is expressed (*Shui et al., 2020*). How innexins and their splice forms contribute to RIM-to-AVA communication, other than requiring *unc-9* function in RIM, remains to be determined.

Interactions between chemical and electrical synapses play prominent roles in motor circuits including the stomatogastric ganglion of crustaceans, the heartbeat circuit in leeches, and rapid escape circuits in nematodes, arthropods, and fish (*Kristan et al., 2005*; *Marder, 1998*; *Szczupak, 2016*). Antagonism between electrical and chemical synapses has also been observed in the reversal-to-omega-turn transition in *C. elegans* (*Wang et al., 2020*). Although chemical synapses in these circuits can be either excitatory or inhibitory, their electrical synapses have all been thought to be

excitatory. We speculate that inhibitory electrical synapses resembling those of RIM gap junctions may emerge as stabilizing elements of other motor circuits with long-lasting, mutually exclusive states.

Optogenetic depolarization of RIM elicits reversals, an effect that is reciprocal to that of RIM hyperpolarization. While gap junctions from RIM to AVA could be attractive candidates for this activity, the overall increase in reversal frequency upon RIM depolarization was only slightly diminished by the *unc-9* gap junction knockdown and unaffected by RIM chemical transmitters. RIM expresses 11 innexin genes, the most of any neuron (*Bhattacharya et al., 2019*). RIM gap junctions may depolarize AVA via innexins that are not affected by the *unc-1(dn)* transgene, such as *inx-1* (*Hori et al., 2018*), which synergizes with *unc-9* to promote reversals evoked by optogenetic RIM depolarization (*Li et al., 2020*). Similarly, the *unc-1(dn)* transgene only partly suppresses the effects of RIM hyperpolarization on reversal frequency, suggesting that *unc-9* may act with additional innexin subunits to propagate RIM hyperpolarization to AVA. Cell-specific knockout of *inx-1* and other innexins should provide a deeper understanding of RIM gap junctions in AVA, AIB, AVE, and other neurons.

### RIM regulates motor state transitions

The dynamic functions of RIM in spontaneous motor state transitions during local search are regulated by the combined action of tyramine, glutamate, and gap junction signaling. All of these synaptic outputs inhibit reversal initiation, even though RIM glutamate and tyramine stabilize the reversal once it has begun.

Among the characterized neurons within the foraging circuit, RIM is the only neuron where ablation has opposite effects on the initiation and execution of a behavioral state (*Gray et al., 2005*). The dynamic transition from forward to backward locomotion requires coordinated activity changes across the circuit, with positive and negative feedback between forward- and reversal-active neurons (*Roberts et al., 2016*; *Figure 1A*). A role for RIM gap junctions in preventing reversals is consistent with its proposed action in the hyperpolarized (forward) state, but tyramine and glutamate release are likely to rely upon depolarization. In one model, a low level of neurotransmitter release during forward-to-reversal transitions might oppose reversal initiation, while higher levels promote it. Low-level release would be consistent with the graded electrical properties of many *C. elegans* neurons, including motor neurons (*Liu et al., 2009*) and RIM (*Liu et al., 2018*), which can also result in graded transmitter release.

We note, however, that chronic or developmental effects of tyramine might also contribute to the increased reversal frequency upon RIM ablation, *tdc-1* mutations, or tetanus toxin expression. Tyramine release during learning can lead to long-term circuit remodeling, and tyramine mediates systemic responses to starvation and other stresses (*De Rosa et al., 2019*; *Ghosh et al., 2016*; *Jin et al., 2016*; *Özbey et al., 2020*). Reversal frequencies during local search are regulated by prior experience on bacterial food, including its density and distribution (*Calhoun et al., 2014*; *López-Cruz et al., 2019*); tyramine is a candidate to mediate this longer-term behavioral effect.

These transitions, as well as the interactions between RIM, AIB, and RIB that promote transitions from reversals to omega turns, deserve fuller scrutiny (*Wang et al., 2020*). Here, we have focused on high-resolution analysis of behavior to complement the increasingly rich studies of neuronal activity associated with locomotion (*Ji et al., 2019*; *Kato et al., 2015*; *Kaplan et al., 2020*; *Nguyen et al., 2016*; *Venkatachalam et al., 2016*). Integration of high-resolution behavior with high-resolution imaging is a critical next step to examine transition dynamics.

## Materials and methods

**Key resources table**

| Reagent type (species) or resource | Designation | Source or reference | Identifiers | Additional information |
|---|---|---|---|---|
| Strain, strain background (*Caenorhabditis elegans* N2, hermaphrodite) | Wild-type | This paper | ID_BargmannDatabase:CX17882 | See *Figures 1–3*, *Figure 2— figure supplements 1 and 2* |

*Continued on next page*

*Continued*

| Reagent type (species) or resource | Designation | Source or reference | Identifiers | Additional information |
|---|---|---|---|---|
| Strain, strain background (*C. elegans* N2, hermaphrodite) | CX0007 | This paper | ID_BargmannDatabase:CX0007 | Child of CX17882; See *Supplementary file 1*, Table 1: Strain details |
| Strain, strain background (*C. elegans* N2, hermaphrodite) | RIM glu KO | This paper | ID_BargmannDatabase:CX17881 | See *Figures 1–3*, *Figure 2—figure supplement 2* |
| Strain, strain background (*C. elegans* N2, hermaphrodite) | *tdc-1* | This paper | ID_BargmannDatabase:CX17883 | See *Figures 1–3*, *Figure 2—figure supplement 2* |
| Strain, strain background (*C. elegans* N2, hermaphrodite) | RIM glu KO; *tdc-1* | This paper | ID_BargmannDatabase:CX17884 | See *Figures 1–3*, *Figure 2—figure supplement 2* |
| Strain, strain background (*C. elegans* N2, hermaphrodite) | *elt-2p*::nGFP; wild-type | This paper | ID_BargmannDatabase:CX18118 | See *Figure 2—figure supplement 1* |
| Strain, strain background (*C. elegans* N2, hermaphrodite) | *elt-2p*::nGFP; edited *eat-4* | This paper | ID_BargmannDatabase:CX17461; ID_BargmannDatabase:CX18118 | See *Figure 2—figure supplement 1* |
| Strain, strain background (*C. elegans* N2, hermaphrodite) | *tbh-1* | DOI:10.1016/j.neuron.2005.02.024 | RRID:SCR_007341: MT9455 | See *Figure 2—figure supplement 2* |
| Strain, strain background (*C. elegans* N2, hermaphrodite) | Wild-type | PMC1213120 | RRID:SCR_007341:N2 | See *Figure 4*, *Figure 4—figure supplements 1 and 2*, *Figure 6—figure supplement 2* |
| Strain, strain background (*C. elegans* N2, hermaphrodite) | RIM::tetanus toxin::mCherry | DOI:10.1016 /j.cell.2015.02.018 | ID_BargmannDatabase:CX14993 | See *Figure 4*, *Figure 4—figure supplements 1 and 2* |
| Strain, strain background (*C. elegans* N2, hermaphrodite) | *tbh-1p*::tetanus toxin::mCherry | This paper | ID_BargmannDatabase:CX17912 | See *Figure 4—figure supplement 1* |
| Strain, strain background (*C. elegans* N2, hermaphrodite) | RIM::HisCl; wild-type | DOI:10.1073/pnas.1400615111 | ID_BargmannDatabase:CX18193 | See *Figures 5 and 7*, *Figure 5—figure supplements 1 and 2* |
| Strain, strain background (*C. elegans* N2, hermaphrodite) | RIM::HisCl; *tdc-1* | This paper | ID_BargmannDatabase:CX18194 | See *Figure 5—figure supplement 1* |
| Strain, strain background (*C. elegans* N2, hermaphrodite) | AVA::GCaMP5; wild-type | DOI:10.1016 j.cell.2015.02.018 | ID_BargmannDatabase: CX15380 | See *Figure 5—figure supplement 3* |
| Strain, strain background (*C. elegans* N2, hermaphrodite) | AVA::GCaMP5; RIM::HisCl | This paper | ID_BargmannDatabase: CX15380; ID_BargmannDatabase: CX18193 | See *Figure 5—figure supplement 3* |

*Continued on next page*

*Continued*

| Reagent type (species) or resource | Designation | Source or reference | Identifiers | Additional information |
|---|---|---|---|---|
| Strain, strain background (*C. elegans* N2, hermaphrodite) | Wild-type | This paper | ID_BargmannDatabase:CX17546 | See *Figure 6*, *Figure 4—figure supplement 1*, *Figure 6—figure supplements 1–3* |
| Strain, strain background (*C. elegans* N2, hermaphrodite) | RIM gap junction KD | This paper | ID_BargmannDatabase:CX14853 | See *Figure 6*, *Figure 6—figure supplements 1–3* |
| Strain, strain background (*C. elegans* N2, hermaphrodite) | *tdc-1* | DOI:10.1016/j.neuron.2005.02.024 | RRID:SCR_007341:MT13113 | See *Figure 6—figure supplement 2* |
| Strain, strain background (*C. elegans* N2, hermaphrodite) | RIM gap junction KD; *tdc-1* | This paper | ID_BargmannDatabase:CX14853 RRID:SCR_007341:MT13113 | See *Figure 6—figure supplement 2* |
| Strain, strain background (*C. elegans* N2, hermaphrodite) | RIC gap junction KD | This paper | ID_BargmannDatabase: CX18293 | See *Figure 6—figure supplement 3* |
| Strain, strain background (*C. elegans* N2, hermaphrodite) | RIM::HisCl; RIM gap junction KD | This paper | ID_BargmannDatabase:CX18137 | See *Figure 7* |
| Strain, strain background (*C. elegans* N2, hermaphrodite) | RIM::ReaChR: wild-type | This paper | ID_BargmannDatabase:CX17885 | See *Figure 7*, *Figure 7—figure supplement 1* |
| Strain, strain background (*C. elegans* N2, hermaphrodite) | RIM::ReaChR: RIM glu KO | This paper | ID_BargmannDatabase:CX17886 | See *Figure 7*, *Figure 7—figure supplement 1* |
| Strain, strain background (*C. elegans* N2, hermaphrodite) | RIM::ReaChR: *tdc-1* | This paper | ID_BargmannDatabase:CX17887 | See *Figure 7*, *Figure 7—figure supplement 1* |
| Strain, strain background (*C. elegans* N2, hermaphrodite) | RIM::ReaChR: RIM glu KO; *tdc-1* | This paper | ID_BargmannDatabase:CX17888 | See *Figure 7*, *Figure 7—figure supplement 1* |
| Strain, strain background (*C. elegans* N2, hermaphrodite) | RIM::ReaChR: wild-type | DOI: 10.1016/j.neuron.2019.01.053 | ID_BargmannDatabase:CX17694 | See *Figure 7*, *Figure 7—figure supplement 1* |
| Strain, strain background (*C. elegans* N2, hermaphrodite) | RIM::ReaChR: RIM gap junction KD | This paper | ID_BargmannDatabase:CX18195 | See *Figure 7*, *Figure 7—figure supplement 1* |
| Chemical compound, drug | Histamine dihydrochloride | Sigma | H7250 | CAS 56-92-8 |
| Chemical compound, drug | (-)-Levamisole hydrochloride | Sigma | L9756 | CAS 16595-80-5 |
| Chemical compound, drug | Polydimethylsiloxane (PDMS) | Sigma | 761036 | 9:1 base:curing agent, Sylgard 184 |
| Software, algorithm | ImageJ | ImageJ (http://imagej.nih.gov/ij/) | RRID:SCR_003070 | Version 1.50i |

*Continued on next page*

*Continued*

| Reagent type (species) or resource | Designation | Source or reference | Identifiers | Additional information |
|---|---|---|---|---|
| Software, algorithm | GraphPad Prism | GraphPad Prism (https://graphpad.com) | RRID:SCR_002798 | Versions 7.0c, 8.4.1 |
| Software, algorithm | MATLAB | MathWorks (https://www.mathworks.com/) | RRID:SCR_001622 | Versions R2014a, R2016b, R2018b |
| Software, algorithm | Metamorph | Molecular Devices (https://www.moleculardevices.com) | RRID:SCR_002368 | Version 7.8.2.0 |
| Software, algorithm | Streampix | Norpix (https://www.norpix.com/products/streampix/streampix.php) | RRID:SCR_015773 | Versions 6 and 8 |
| Software, algorithm | Python | Python (https://www.python.org/) | RRID:SCR_008394 | Version 3.8.1 |
| Software, algorithm | Analysis code | This paper (https://doi.org/10.5061/dryad.ht76hdrf6, https://github.com/navinpokala/BargmannWormTracker) | | See Dryad repository or Github |

## Nematode and bacterial culture

In all experiments, bacterial food was *E. coli* strain OP50. Nematodes were grown at room temperature (21–22°C) or at 20°C on nematode growth media plates (NGM; 51.3 mM NaCl, 1.7% agar, 0.25% peptone, 1 mM $CaCl_2$, 12.9 µM cholesterol, 1 mM $MgSO_4$, 25 mM $KPO_4$, pH 6) seeded with 200 µL of a saturated *E. coli* liquid culture grown in LB at room temperature or at 37°C, and stored at 4°C (*Brenner, 1974*). All experiments were performed on young adult hermaphrodites, picked as L4 larvae the evening before an experiment.

WT controls are derived from the N2 Bristol strain, and an additional WT strain CX0007 was derived by loss of the transgene from CX17882, to maximize the similarity of controls within an experiment. Mutant strains were backcrossed into N2 at least 3× to reduce background mutations. WT controls in all figures were matched to test strains for transgenes and co-injection markers. Full genotypes and detailed descriptions of all strains and transgenes appear in *Supplementary file 1*, Table 1:. Strain details.

## Molecular biology and transgenics

A 4.5 kb region upstream of *tdc-1* that drives expression in RIM and RIC neurons was used for all RIM manipulations. In all cases other than the RIM glutamate knockout, these reagents affect RIC as well as RIM. To separate the functions of the RIM and RIC neurons, we used a 4.5 kb region upstream of *tbh-1* to drive expression in RIC neurons. Phenotypes specific to the *tdc-1* transgenes were inferred to have an essential contribution from RIM. Relevant strains and plasmids are described in *Supplementary file 1*, Table 1: Strain details, and *Supplementary file 1*, Table 2:. Plasmids generated for this study.

Transgenic animals were generated by microinjecting the relevant plasmid of interest with a fluorescent co-injection marker (*myo-2p*::mCherry, *myo-3p*::mCherry, *elt-2p*::nGFP, *elt-2p*::mCherry, *unc-122p*::GFP) and empty pSM vector to reach a final DNA concentration of 100 ng/µL. Transgenes were maintained as extrachromosomal arrays.

## Foraging assay

Off-food foraging assays were performed and analyzed following *López-Cruz et al., 2019*. Animals were first preconditioned to a homogenous *E. coli* lawn to standardize their behavioral state (*Calhoun et al., 2014*). The homogenous lawn was made by seeding NGM plates with a thin uniform layer of saturated *E. coli* liquid culture ~16 hr before the beginning of the assay. 20 young adult hermaphrodites were placed on this lawn for 45 min prior to recording and constrained to a fixed area of 25 cm² using filter paper soaked in 20 mM $CuCl_2$. 12–15 of these preconditioned animals were transferred to an unseeded NGM plate briefly to clear adherent bacteria, and then transferred to a large unseeded NGM assay plate, where they were constrained to a fixed area of ~80 cm² using filter paper soaked in 20 mM $CuCl_2$. Video recordings of these animals began within 5–6 min from food removal to capture local search behavior. Animals were recorded for 45 min as previously described using a 15

MP PL-D7715 CMOS video camera (Pixelink). Frames were acquired at 3 fps using Streampix software (Norpix) using four cameras to image four assays in parallel. LED backlights (Metaphase Technologies) and polarization sheets were used to achieve uniform illumination (*López-Cruz et al., 2019*). Each experimental strain was assayed a minimum of six times over 2 days, with control strains assayed in parallel.

Animals were tracked using custom MATLAB software (BargmannWormTracker) without manual correction of tracks (*López-Cruz et al., 2019*; *Pokala et al., 2014*). Tracker software is available at: https://github.com/navinpokala/BargmannWormTracker (*Pokala, 2019*).

## Quantification of spontaneous behavior

Behavioral states were extracted from the State array generated by BargmannWormTracker. Local search event frequencies per minute were calculated 4–8 min after removal from food. Global search frequencies per minute were calculated 36–40 min after removal from food. Only tracks that were continuous for the entire 4 min time interval were included in frequency analysis. When calculating frequencies, tracks taken on a single day from a single assay plate were averaged to give a single data point, for example, in *Figure 2B and D*.

Distributions of reversal parameters and forward run durations were calculated using events observed during local search, 4–8 min after removal from food. All reversals were included; only forward runs over 2 s in length were included. Reversal length is the path length calculated using the X-Y coordinates, worm length, and pixel size extracted from the tracker. Reversal and forward run speed are the average of mean and median speed extracted from the tracker.

Tracks that were less than 5 min long, tracks approaching the copper barrier, and tracks that did not include a complete reversal or forward run were not included in reversal and forward run parameter analyses.

Data and relevant functions pertaining to these analyses are available at Dryad: https://doi.org/10.5061/dryad.ht76hdrf6 and GitHub: https://github.com/BargmannLab/SordilloBargmann2021.

## Optogenetic manipulations

The red-shifted channelrhodopsin ReaChR (Lin et al., 2013) was expressed under the *tdc-1* promoter and animals were stimulated during the off-food foraging assay described above, following *López-Cruz et al., 2019*. Experimental animals were treated with 12.5 µM all-trans retinal overnight and during preconditioning on homogeneous food lawns; control animals were prepared in parallel on plates that did not contain retinal. Optogenetic stimuli were delivered with a 525 nm Solis High-Power LED (ThorLabs) controlled by custom MATLAB software and strobed at a 5% duty cycle. Two (*Figure 7D–E and Figure 7—figure supplement 1A–C*) or three (*Figure 7F and G and Figure 7—figure supplement 1D–F*) pulses of ~45 µW/mm$^2$ light were delivered for 20 s each with a 100 s interpulse interval starting at 8 or 10 min (local search) and 38 or 40 min (global search). These light intensities elicited the maximal behavioral effect of ReaChR. Additional lower light intensities (not shown) were included in each experiment, with pulses always separated by 100 s.

For behavioral quantification, tracks were aligned around the light pulses and extracted over a 120 s period, with the light pulse delivered at 50–70 s. Only tracks that were continuous for the entire 120 s period were used. The change in reversal frequency was calculated by subtracting the mean reversal frequency during an 18 s time window before light onset from the mean reversal frequency during an 18 s time window during the light pulse. Behavioral parameters were scored only for the first reversal of duration ≥0.5 s that began during the light stimulation.

## Acute histamine treatment

The *Drosophila* histamine-gated chloride channel HisCl1 was expressed under the *tdc-1* promoter. Animals were treated with histamine following *Pokala et al., 2014*. Histamine dihydrochloride (Sigma-Aldrich H7250) was dissolved in Milli-Q purified water, filtered, and stored at –20°C. Histamine solution was added to cooled NGM (45–50°C) for a final concentration of 10 mM to make assay plates. Animals were habituated on homogeneous OP50 lawns on histamine-free NGM plates, transferred to food-free, histamine-free NGM plates for cleaning, and then recorded on 10 mM histamine assay plates for 45 min. See 'Foraging assay' and 'Quantification of spontaneous behavior' sections.

## Chronic histamine treatment

Histamine was prepared as above. Treated animals were grown on 10 mM histamine plates seeded with OP50 lawns for ~48 hr prior to being assayed, and were habituated, cleaned, and assayed on 10 mM histamine plates. Untreated controls were grown, transferred, and assayed in parallel on histamine-free NGM plates. Treated and untreated animals were subsequently retrieved from assay plates and transferred to histamine-free NGM plates seeded with a homogeneous OP50 lawn for ~60–90 min to allow treated animals to recover from the histamine treatment. All animals were then assayed a second time on histamine-free plates. See 'Foraging assay' and 'Quantification of spontaneous behavior' sections.

## AVA GCaMP imaging

GCaMP 5.0 was expressed in AVA under the *rig-3* promoter. GCaMP dynamics were imaged in a high-throughput microfluid chip following Dobosiewicz, Liu, and Bargmann, 2019. Before beginning the experiment, animals were removed from food and gently washed in NGM buffer (51.3 mM NaCl, 0.25% peptone, 1 mM $CaCl_2$, 1 mM $MgSO_4$, 25 mM $KPO_4$). Approximately 20 animals of each genotype were then loaded into separate arenas of a custom-fabricated two-arena polydimethylsiloxane (PDMS; Sigma 761036, made from 9:1 base:curing agent, Sylgard 184) imaging device. Conditional media were prepared by inoculating NGM buffer with a single colony of OP50 bacteria, incubating overnight in a 37°C shaking incubator (final OD600 = 0.3–0.4), and removing bacteria with 0.22 μm filters (Millex). Animals were paralyzed for ~50 min in darkness in conditioned media with 2 mM levamisole (Sigma) and 10 mM histamine. Conditioned media were replaced with NGM buffer with 2 mM levamisole and 10 mM histamine 5 min after the recording began to evoke a local search-like state. GCaMP dynamics were imaged at 10 frames/s for 40 min and tracked using custom ImageJ software. Two experiments were performed over 2 days.

Experiments were performed on a Zeiss AxioObserver A1 inverted microscope fit with a 5×/ 0.25 NA Zeiss Fluar objective, a Hamamatsu Orca Flash 4 sCMOS camera with a 0.63× c-mount adapter to increase field of view. 474 nm wavelength light was delivered with a Lumencor SOLA-LE lamp. Metamorph 7.8.2. software was used to control image acquisition, light pulsing, stimulus switching (National Instruments NI-DAQmx connected to an Automate Valvebank 8 II actuator that controls a solenoid valve), and stimulus selection (Hamilton 8-way distribution valve).

## AVA GCaMP analysis

Spontaneous AVA GCaMP dynamics were analyzed at 0–5 min after conditioned media removal (local search) and 30–35 min after conditioned media removal (global search). 15–20 tracks were analyzed from each genotype per experiment.

ON and OFF states were determined using methods adapted from *Gordus et al., 2015*. Custom Python and MATLAB scripts were used to quantify fluorescence in AVA. Data were smoothed over 1 s (10 frame) intervals. The median 10% of the lowest observed fluorescence was set as $F_0$ and used to calculate the change of fluorescence for each frame ($\Delta F = F\ F_0$), which was subsequently normalized to $F_0$ ($\Delta F/F_0$). Smoothed traces with $\Delta F/F_0$ >10% were given an initial binary ON state assignment defined as above or below 50% of the Fmax. Subsequently, (1) the time derivative (dF/dt) for each trace was calculated and smoothed over 3 s intervals, (2) threshold dF/dt parameters for ON/OFF transitions were defined based on minima and maxima of each dF/dt and (3) final ON/OFF states and were defined using both the dF/dt parameters and the initial binary assignment. Multiple thresholds and correction factors were tested; while they led to small changes in absolute values, they did not affect the conclusions about effects of RIM silencing on AVA ON and OFF states.

Relevant functions pertaining to these analyses are available at Dryad: https://doi.org/10.5061/dryad.ht76hdrf6 and GitHub: https://github.com/BargmannLab/SordilloBargmann2021.

## Statistical analyses

All statistical analyses were conducted in GraphPad Prism except for the two-sample Kolmogorov–Smirnov test, which was performed in MATLAB. When making multiple comparisons, the p-values of the two-sample Kolmogorov–Smirnov test were adjusted with a Bonferroni correction. The effect size was calculated for all significant distribution comparisons as the D statistic, which represents the maximum distance between the empirical cumulative distributions of the data. Because of the large n

values in these experiments, even very small effects reached statistical significance. Based on control strains (e.g., *Figure 2—figure supplement 1*), we set a meaningful effect size of ≥0.15 as a cutoff for discussing results. See *Supplementary file 1*, Table 3: Reversals and forward runs, n values. A summary of all p-values and statistical tests can be found in *Supplementary file 1*, Table 4: Statistical analysis. Sample sizes and experimental design were selected based on previous experiments that used the same assays and similar perturbations in *López-Cruz et al., 2019*.

## Acknowledgements

We thank Philip Kidd, Andrew Gordus, Qiang Liu, Elias Scheer, Javier Marquina-Solis, Audrey Harnagel, James Lee, Friederike Buck, Leslie Vosshall, Vanessa Ruta, Yishi Jin, and Jeremy Dittman for thoughtful discussions and comments on the manuscript. We thank Audrey Harnagel for her help in designing and executing the AVA imaging experiment. We thank Alejandro López-Cruz for his collaboration in creating the cell-specific *eat-4* knockout strain. This work was supported by the Howard Hughes Medical Institute, of which CIB was an investigator, and by the Chan Zuckerberg Initiative.

## Additional information

### Funding

| Funder | Grant reference number | Author |
| --- | --- | --- |
| Howard Hughes Medical Institute | CIB was an HHMI Investigator | Cornelia I Bargmann Aylesse Sordillo |
| Chan Zuckerberg Initiative | Lab support to CIB | Aylesse Sordillo |

The funders had no role in study design, data collection and interpretation, or the decision to submit the work for publication.

### Author contributions

Aylesse Sordillo, Conceptualization, Data curation, Formal analysis, Investigation, Methodology, Visualization, Writing - original draft, Writing - review and editing; Cornelia I Bargmann, Conceptualization, Funding acquisition, Supervision, Writing - review and editing

### Author ORCIDs

Aylesse Sordillo (iD) http://orcid.org/0000-0002-6678-8748
Cornelia I Bargmann (iD) http://orcid.org/0000-0002-8484-0618

### Decision letter and Author response

Decision letter https://doi.org/10.7554/eLife.67723.sa1
Author response https://doi.org/10.7554/eLife.67723.sa2

## Additional files

### Supplementary files
• Transparent reporting form

• Source data 1. Source data for all dot plots. Includes numerical values representing the average frequency of a behavioral event, per animal, per minute, on a single assay plate.

• Source data 2. Source data for *Figure 5—figure supplement 3*.

• Supplementary file 1. Supplementary Tables 1-4.

### Data availability

All primary behavioral data and relevant functions pertaining to data analysis are available at Dryad (https://doi.org/10.5061/dryad.ht76hdrf6) and Github (https://github.com/BargmannLab/Sordillo-Bargmann2021; copy archived at swh:1:rev:d528552991e834f6aa5d7d6dde63ec23e799fc93). Source data 1 includes raw numbers for all dot plots in Figures 2-7 and supplementary figures. Source data

2 includes raw numbers for Figure 5—figure supplement 3. Tracker software is available at: https://github.com/navinpokala/BargmannWormTracker.

The following dataset was generated:

| Author(s) | Year | Dataset title | Dataset URL | Database and Identifier |
|---|---|---|---|---|
| Sordillo A, Bargmann CI | 2021 | Behavioral control by depolarized and hyperpolarized states of an integrating neuron | https://doi.org/10.5061/dryad.ht76hdrf6 | Dryad Digital, 10.5061/dryad.ht76hdrf6 |

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
