## [Decision Letter]

**Acceptance summary:**

In this work, authors focused on the role of RIM neurons, which are known to be important for locomotion control of *C. elegans*, but whose precise role was unclear and enigmatic. Now they show that RIM depolarization extends reversals via synaptic (glutamatergic) and secretory (tyraminergic) signaling, while RIM hyperpolarization promotes forward locomotion via electrical signaling through gap junctions. As a result, RIM was shown to act for stabilizing both forward and backward movement, which is important for understanding of *C. elegans* behavior in general. Upon revision, authors added new experimental results and edited the text to add explanation to clarify the ambiguity.

**Decision letter after peer review:**

Thank you for submitting your article "Behavioral control by depolarized and hyperpolarized states of an integrating neuron" for consideration by *eLife*. Your article has been reviewed by 3 peer reviewers, including Yuichi Iino as Reviewing Editor and Reviewer #1, and the evaluation has been overseen by Ronald Calabrese as the Senior Editor.

Essential revisions:

1) As pointed out by reviewer #2 point 1, evaluation of the effect of manipulation of RIM activity, especially by HisCl, on connected neurons needs to be performed, probably by calcium imaging. A similar test was previously done by Gordus et al., 2015, Figure 4, in odor-stimulated animals. Authors need to do similar experiments but in the absensce of stimuli and in the local and global off-food state, to see how it effects spontaneous rather than evoked ON/OFF state transitions in AIB and AVA, for example.

2) As pointed out by reviewer #3 point 2, it is essential to clarify the difference between RIM ablation and RIM transient silencing. The HisCl silencing differs from ablation in two ways. One is acute v. chronic perturbation, and the other is hyperpolarization vs. absence of a cell. The authors focus on the latter, but interpretation of the data would change if they found that chronic hyperpolarization did not cause the same effect as acute hyperpolarization, such as compensatory changes in the neuronal functions.

3) As pointed by reviewer #1, authors' claim that activation by RIM depolarization is through synaptic transmission and inhibition by RIM hyperpolarization is through gap junctions needs to be strengthened. RIM::HisCl prolongs forward movement and reduces reversal rate both in the tdc-1 (Figure 5C) and gap junction knockdown background (Figure 7B), but the his- data in Figure 5C look different from Figure 2B. Authors also claim that reversal length is shortened in the tdc-1/eat-4 mutants (Figure 3), but not in the gap junction knockdown mutant (Figure 6E). However, the wild type data differ considerably between the two figures. More repeated measurements are needed to build confidence in the authors' interpretation of data given that the baselines wander.

Authors are encouraged to address other comments/questions by the reviewers, such as, why the local/global difference? Why would hyperpolarization propagate more readily than depolarization through the gap junctions?

*Reviewer #1 (Recommendations for the authors):*

My only concern is that it is still not clear whether the HisCl effect is mediated by gap junctions, because HisCl and unc-1(n494) have just additive effect causing mutual suppression. This is expected to happen also in cases where HisCl and unc-1 act in different pathways. The same ambiguity remains for the ReaChR experiments. Although RIM expresses many gap junction genes and the specificity of unc-1(n494) is not known, inx-1 is a good candidate, because previous study (Hori et al., 2018) suggested that inx-1 is not down regulated by unc-1(n494). inx-1 mutants exist and the defect in locomotion is mild. It is recommended that authors test inx-1 mutant, or inx-1; unc-1(n494), and test the effect of HisCl and ReaChR under these backgrounds. It is noted, however, that considering 11 innexins are expressed in RIM, there is no guarantee that this experiment will turn out informative.

In figure 2, data show that tdc-1 mutants have higher frequency of reversal, while this effect was not observed in Figure 5C (without His). Is there any speculation by the authors about the cause of this discrepancy?

*Reviewer #2 (Recommendations for the authors):*

1. Physiological effects are not measured in this study, which is a little surprising given the availability of tools. This might be particularly interesting in the experiments related to electrical coupling (HisCl and innexin dominant negative), as it would strengthen the claim that RIM electrical synapses act to suppress activity in reversal command neurons to prolong the forward state. That said, the conclusions stand on the interpretation of the behavioural results.

2. It is surprising that the electrical coupling between RIM and the reversal circuit appears to play only an inhibitory role. I would have expected the dnUNC-1 experiment to destabilize both forward and reverse states, but this does not appear to be the case, as reversal measurements (speed, length, duration) are unchanged. And then there is the puzzling and weak/differential effect (lines 265-267) of dnUNC-1 in local vs global search. Finally it seems odd that optogenetically induced but not spontaneous reversals are destabilized by disrupting RIM gap junction signaling. I wonder if the authors have thoughts on why this relationship seems to have this directional bias, and what might difference between spontaneous and evoked reversals might explain this.

3. Figure 2 and 3. Much of the analysis uses a distinction between "long" and "short" reversals. This distinction is based on a clearly bimodal distribution (Figure 2C) that supports the idea that these are 2 categorically different behaviors with overlapping length distributions. The use of a cutoff will miscategorize some "long" distributions at the short end of that distribution as "short" and vice versa, but that's fine. What's less clear is that using this cutoff makes sense in the case of some of the mutants/transgenics analyzed. In Figure 2E it appears that in tdc-1 and tdc-1; RIM-Glu-KO animals, long reversals are essentially abolished-the bimodal shape of the WT and RIM-Glu-KO distributions is gone, and the reversals that fall into the "long" category by virtue of being above the WT-based cutoff are the tail of the short reversal distribution (which is masked by the overlap in WT). This seems to make sense with the rev-omega reduction seen when tdc-1 is mutated. I can't tell, but is it plausible that the differences in reversal-omega is sufficiently explained by the fact that all tdc-1 reversals (no matter which side of the cutoff they are on) are behaviorally "short reversals"? I don't think this raises concerns about the major conclusions of the paper, but I am wondering if there is an analysis approach that considers the fit of the data to one or two distributions, and whether it might simplify the interpretation of some of the

analysis.

4. Figure 4. With the above in mind, seeing the full distribution of reversal lengths in Figure 4 would be instructive.

*Reviewer #3 (Recommendations for the authors):*

The manuscript sets the stage for two potentially exciting discoveries. The first is that RIMs use tyraminergic and glutamatergic co-transmission to control foraging. Because VGLUT is intertwined with monoamine signaling, this point demands another experiment to conclusively demonstrate that glutamate from RIMs is important for RIM control of foraging. The second is that RIMs have separable functions when depolarized and hyperpolarized. The model presented predicts that chronic hyperpolarization will have effects similar to acute hyperpolarization. In addition to testing the model, it will build confidence to see that two different methods to hyperpolarize RIMs cause similar effects on foraging behavior.

---

## [Author Response]

Essential revisions:1) As pointed out by reviewer #2 point 1, evaluation of the effect of manipulation of RIM activity, especially by HisCl, on connected neurons needs to be performed, probably by calcium imaging. A similar test was previously done by Gordus et al., 2015, Figure 4, in odor-stimulated animals. Authors need to do similar experiments but in the absensce of stimuli and in the local and global off-food state, to see how it effects spontaneous rather than evoked ON/OFF state transitions in AIB and AVA, for example.

We addressed this point with a new set of experiments added as Figure 5- supplement 3 and discussed on page 10 of the text. As requested, we imaged spontaneous calcium transients in AVA neurons in RIM::HisCl animals, which were immobilized in a configuration similar to the one in the Gordus paper referenced by the reviewer. We used immobilized animals because the results are quantitatively reproducible, and we and others have experience in conducting and interpreting them.

High AVA activity is a proxy for the reversal state. To generate a state analogous to local search, we switched the immobilized animals from bacteria-conditioned medium to buffer. Over a 30-minute period in buffer, the fraction of time animals spent in a high AVA activity state decreased, as predicted if features of local and global search are represented in these animals.

Hyperpolarizing RIM with RIM::HisCl led to the following changes (1) during the “local search” interval, the fraction of time animals spent in a high AVA activity state was reduced to the level appropriate for “global search” (2) during “local search”, the number of transitions from a low to a high AVA activity state was reduced. These results are consistent with the behavioral results indicating that RIM::HisCl reduces the duration and frequency of reversals during local search (Figure 5A-C). Notably, AVA activity in the “global search” period was less affected, in agreement with the behavioral effects of RIM::HisCl.

We did not examine AIB calcium responses, because AIB activity is often uncoupled from that of RIM and AVA, and is correlated with turning states as well as reversals (Gordus et al., 2015; Wang et al., 2020), so it would be difficult to interpret these results.

In summary, these imaging results support our behavioral results and suggest that silencing RIM suppresses reversals by reducing AVA activity. However, it is worth noting that immobilization alters neuronal dynamics in *C. elegans* (Hallinen et al., 2021). Imaging freely-moving animals would be a more definitive experiment, but those methods are still in development (in a number of labs), and establishing and validating them is beyond the scope of this paper. We include a statement of this limitation in the revised paper (p. 10).

2) As pointed out by reviewer #3 point 2, it is essential to clarify the difference between RIM ablation and RIM transient silencing. The HisCl silencing differs from ablation in two ways. One is acute v. chronic perturbation, and the other is hyperpolarization vs. absence of a cell. The authors focus on the latter, but interpretation of the data would change if they found that chronic hyperpolarization did not cause the same effect as acute hyperpolarization, such as compensatory changes in the neuronal functions.

As the reviewer notes, there can be differences between acute and chronic neuronal inactivation. Indeed, a recent paper identifies a *C. elegans* sensory circuit in which chronic silencing results in functional compensation, masking the effects of acute silencing (Yeon et al., 2021). To address this point, we conducted experiments in which we chronically silenced RIM for 48 hours using the RIM::HisCl system, and then tested the animals for behavior. We observed very similar effects of chronic and acute RIM silencing on local search behavior, including the unexpected decrease in reversal frequency as well as the expected decrease in reversal duration (now added as Figure 5F-5H). These results support the interpretation that RIM silencing is different from RIM ablation.

3) As pointed by reviewer #1, authors' claim that activation by RIM depolarization is through synaptic transmission and inhibition by RIM hyperpolarization is through gap junctions needs to be strengthened. RIM::HisCl prolongs forward movement and reduces reversal rate both in the tdc-1 (Figure 5C) and gap junction knockdown background (Figure 7B), but the his- data in Figure 5C look different from Figure 2B. Authors also claim that reversal length is shortened in the tdc-1/eat-4 mutants (Figure 3), but not in the gap junction knockdown mutant (Figure 6E). However, the wild type data differ considerably between the two figures. More repeated measurements are needed to build confidence in the authors' interpretation of data given that the baselines wander.

Technical issue: As noted by Reviewer #3, we conducted all experiments together with controls that were matched for genetic background and co-injected plasmids. Different wild-type control strains were used in Figures 2 and 3 (*tdc-1p*::*nFLP, elt-2p*::nGFP*)* versus Figure 5 and 7 (*tdc-1p*::HisCl::SL2::mCherry*)* and Figure 6 *(unc-122p*::GFP*)*, as described in Supplementary File 1, Table S1, strain details. These differences, plus small day-to-day variation between experiments (also noted in López-Cruz et al., 2019 and Zhao et al., 2003), are the likely source of the differences in baseline that Reviewer #1 noted. We clarify that point on page 5 of the text.

We performed additional repetitions of the acute histamine silencing experiments, and obtained the same results (i.e. Figures 5C-E and 7B-C now include more replicates). Additionally, the new Figure 5 includes a chronic silencing experiment, which repeats the results of acute silencing and also includes recovery after removal from histamine (Figure 5—figure supplement 2).

Interpretation/discussion, Reversal length in Figure 3 vs Figure 6E: Comparing reversal measures, there is, in fact, a small decrease in reversal length in the gap junction knockdown (0.11), which narrowly missed our threshold for discussion (0.15) but was smaller than the decrease in synaptic mutants (0.20-0.46). The effect of the gap junction knockdown was repeated in separate experiments in Figure 6 —figure supplement 2, with a slightly larger effect (0.16), and was additive with the synaptic mutants. In addition, reversals induced by optogenetic activation of RIM are shorter in the gap junction knockdown (Figure 7G). We changed the language on page 11 to indicate that the gap junction knockdown affected reversal length, but to a lesser extent than synaptic mutants.

Authors are encouraged to address other comments/questions by the reviewers, such as, why the local/global difference? Why would hyperpolarization propagate more readily than depolarization through the gap junctions?Reviewer #1 (Recommendations for the authors):My only concern is that it is still not clear whether the HisCl effect is mediated by gap junctions, because HisCl and unc-1(n494) have just additive effect causing mutual suppression. This is expected to happen also in cases where HisCl and unc-1 act in different pathways. The same ambiguity remains for the ReaChR experiments. Although RIM expresses many gap junction genes and the specificity of unc-1(n494) is not known, inx-1 is a good candidate, because previous study (Hori et al., 2018) suggested that inx-1 is not down regulated by unc-1(n494). inx-1 mutants exist and the defect in locomotion is mild. It is recommended that authors test inx-1 mutant, or inx-1; unc-1(n494), and test the effect of HisCl and ReaChR under these backgrounds. It is noted, however, that considering 11 innexins are expressed in RIM, there is no guarantee that this experiment will turn out informative.

Additive effects: We know that the *unc-1(dn)* does not inactivate all gap junctions, and believe that is the reason for the partial effect of the knockdown in Figure 7, either alone (Figure 7F-G) or with RIM::HisCl (Figure 7B-C). We agree that *inx-1* is a strong candidate to mediate the remaining effect. In a recent paper, Li et al., (2020) showed that *inx-1; unc-9* double mutants are defective in reversal initiation upon acute RIM depolarization, but single mutants are not. These results match our single mutant knockdown in Figure 7F, and support the model that *inx-1* (*unc-1(dn)*-resistant) and *unc-9 (unc-1(dn)-*sensitive) gap junctions both propagate depolarizing RIM signals to drive reversals. At the same time, the complementary value of our experiment is the ability to manipulate one neuron, RIM, as most gap junction subunits are expressed in many cells and could be affecting multiple parts of the circuit. We currently lack reagents for cell-specific knockout of *inx-1.* We now clarify these points on page 16 and cite the Li paper for the possible contribution of *inx-1.*

In figure 2, data show that tdc-1 mutants have higher frequency of reversal, while this effect was not observed in Figure 5C (without His). Is there any speculation by the authors about the cause of this discrepancy?

We acknowledge the concern with Figure 5C; in addition to our Figure 2, previous reports from Alkema at al., 2005 and Li et al., 2020 also show higher frequency of reversal in *tdc-1* mutants. We do not know why this experiment is anomalous, and have moved it to Figure 5 —figure supplement 1.

Reviewer #2 (Recommendations for the authors):1. Physiological effects are not measured in this study, which is a little surprising given the availability of tools. This might be particularly interesting in the experiments related to electrical coupling (HisCl and innexin dominant negative), as it would strengthen the claim that RIM electrical synapses act to suppress activity in reversal command neurons to prolong the forward state. That said, the conclusions stand on the interpretation of the behavioural results.

See Essential Revision #1. Although this reviewer said that behavioral results were sufficient, we recognize the issue and imaged spontaneous calcium transients in AVA command neurons in RIM::HisCl animals. We obtained results indicating that silencing RIM reduces AVA activity, now included in Figure 5 —figure supplement 3 and described on page 10.

2. It is surprising that the electrical coupling between RIM and the reversal circuit appears to play only an inhibitory role. I would have expected the dnUNC-1 experiment to destabilize both forward and reverse states, but this does not appear to be the case, as reversal measurements (speed, length, duration) are unchanged. And then there is the puzzling and weak/differential effect (lines 265-267) of dnUNC-1 in local vs global search. Finally it seems odd that optogenetically induced but not spontaneous reversals are destabilized by disrupting RIM gap junction signaling. I wonder if the authors have thoughts on why this relationship seems to have this directional bias, and what might difference between spontaneous and evoked reversals might explain this.

See Essential Revision #3. Why aren’t reversal states shorter in the gap junction knockdown? In fact, there was a small decrease in reversal length in the gap junction knockdown (0.11), which narrowly missed our threshold for discussion (0.15) but was smaller than the decrease in synaptic mutants (0.20-0.46). The result was reproducible in a different strain background, where the effect size was slightly greater (0.16; Figure 6 —figure supplement 2). That matches the observation that the reversals induced by optogenetic activation of RIM are shorter in the gap junction knockdown (Figure 7G). We changed the language on page 11 to say that the effect of the gap junction knockdown on reversal length was less than that of the synaptic mutants. We also note that RIM expresses genes for 11 innexins, and it is possible that other innexins propagate depolarizing currents to a greater extent than UNC-9. For example, *inx-1* is resistant to the *unc-1(n494)* dominant negative transgene (Hori et al., 2018) and acts in parallel with *unc-9* to propagate RIM depolarization (Li et al., 2020). We now make these points on page 16.

As the reviewer notes, we see evidence that RIM gap junctions propagate depolarizing currents in the optogenetic experiments in Figure 7F-G. First, reversals are invoked less efficiently during global search in *unc-1(dn)* gap junction knockdowns, and second, all evoked reversals are shorter. At this point we can only speculate on the difference between spontaneous and evoked reversals, and local versus global search reversals.

3. Figure 2 and 3. Much of the analysis uses a distinction between "long" and "short" reversals. This distinction is based on a clearly bimodal distribution (Figure 2C) that supports the idea that these are 2 categorically different behaviors with overlapping length distributions. The use of a cutoff will miscategorize some "long" distributions at the short end of that distribution as "short" and vice versa, but that's fine. What's less clear is that using this cutoff makes sense in the case of some of the mutants/transgenics analyzed. In Figure 2E it appears that in tdc-1 and tdc-1; RIM-Glu-KO animals, long reversals are essentially abolished-the bimodal shape of the WT and RIM-Glu-KO distributions is gone, and the reversals that fall into the "long" category by virtue of being above the WT-based cutoff are the tail of the short reversal distribution (which is masked by the overlap in WT). This seems to make sense with the rev-omega reduction seen when tdc-1 is mutated. I can't tell, but is it plausible that the differences in reversal-omega is sufficiently explained by the fact that all tdc-1 reversals (no matter which side of the cutoff they are on) are behaviorally "short reversals"? I don't think this raises concerns about the major conclusions of the paper, but I am wondering if there is an analysis approach that considers the fit of the data to one or two distributions, and whether it might simplify the interpretation of some of the analysis.

We agree that the reversals in *tdc-1* may all be “short.” Recent work from Wang et al., (2020) has elegantly shown different kinetics and circuit mechanisms for short reversals (not coupled to turns) and long reversals (coupled to turns). While those reversals were evoked, rather than spontaneous, their general behavioral results match our results in Figures 2-3. They studied a different set of cells and mutants from those described here, so the results are complementary to ours. We cite those results on page 7.

4. Figure 4. With the above in mind, seeing the full distribution of reversal lengths in Figure 4 would be instructive.

Figure 4 (RIM::Tetanus toxin) is a good confirmation of the overall synaptic results, but we would like to focus on figures that drive our main conclusions. The primary data are all available on Dryad and Github for use by any interested scientist.

Reviewer #3 (Recommendations for the authors):The manuscript sets the stage for two potentially exciting discoveries. The first is that RIMs use tyraminergic and glutamatergic co-transmission to control foraging. Because VGLUT is intertwined with monoamine signaling, this point demands another experiment to conclusively demonstrate that glutamate from RIMs is important for RIM control of foraging. The second is that RIMs have separable functions when depolarized and hyperpolarized. The model presented predicts that chronic hyperpolarization will have effects similar to acute hyperpolarization. In addition to testing the model, it will build confidence to see that two different methods to hyperpolarize RIMs cause similar effects on foraging behavior.

With respect to glutamatergic transmission, Li et al. (2020) have identified a glutamate receptor subunit, *avr-14,* that affects spontaneous reversals and shows a genetic interaction with RNAi knockdown of *eat-4* in RIM. These results support the suggestion that RIM uses glutamate as a transmitter. We now cite that result on page 14.

With respect to acute and chronic silencing, see Essential Revisions point 2. We added new data to Figure 5F-H showing that similar behavioral results upon acute or chronic (48 hours) silencing of RIM. We also show rapid recovery from chronic silencing in Figure 5 —figure supplement 2.

References

Gordus, A., Pokala, N., Levy, S., Flavell, S. W., and Bargmann, C. I. (2015). Feedback from network states generates variability in a probabilistic olfactory circuit. Cell, 161(2), 215-227. doi:10.1016/j.cell.2015.02.018

Hallinen, K. M., Dempsey, R., Scholz, M., Yu, X., Linder, A., Randi, F., et al., (2021). Decoding locomotion from population neural activity in moving *C. elegans*. eLife, 10. doi:10.7554/eLife.66135

Hori, S., Oda, S., Suehiro, Y., Iino, Y., and Mitani, S. (2018). Off-responses of interneurons optimize avoidance behaviors depending on stimulus strength via electrical synapses. PLoS Genet, 14(6), e1007477. doi:10.1371/journal.pgen.1007477

Li, Z., Zhou, J., Wani, K., Yu, T., Ronan, E. A., Piggott, B. J., et al., (2020). A *C. elegans* neuron both promotes and suppresses motor behavior to fine tune motor output. BioRxiv, 11.02.354472. doi:10.1101/2020.11.02.354472

Lopez-Cruz, A., Sordillo, A., Pokala, N., Liu, Q., McGrath, P. T., and Bargmann, C. I. (2019). Parallel multimodal circuits control an innate foraging behavior. Neuron, 102(2), 407-419 e408. doi:10.1016/j.neuron.2019.01.053

Wang, Y., Zhang, X., Xin, Q., Hung, W., Florman, J., Huo, J., et al., (2020). Flexible motor sequence generation during stereotyped escape responses. eLife, 9. doi:10.7554/eLife.56942

Yeon, J., Takeishi, A., and Sengupta, P. (2021). Chronic vs acute manipulations reveal degeneracy in a thermosensory neuron network. MicroPubl Biol, 2021. doi:10.17912/micropub.biology.000355

Zhao, B., Khare, P., Feldman, L., and Dent, J. A. (2003). Reversal frequency in *Caenorhabditis elegans* represents an integrated response to the state of the animal and its environment. J Neurosci, 23(12), 5319-5328. PMC6741178